# Dynamic High-Dimensional Facility Location with Low Recourse

**Sayan Bhattacharya** [1]  **Martín Costa** [2]  **Silvio Lattanzi** [2]  **Jakub Łącki** [2]  **Nikos Parotsidis** [2]

## Abstract

We study the problem of dynamic facility location with non-uniform costs. Facility location is a central problem in unsupervised learning and in recent years the dynamic version of the problem has been extensively studied. In this paper, we study the setting where clients are added and deleted in real-time and one is interested in maintaining efficiently a stable and high-quality solution. Interestingly, we are able to show that on High Dimensional Euclidean metrics it is possible to obtain efficient algorithms for this problem. More formally, we obtain a randomized algorithm for dynamic facility location in $d$-dimensional Euclidean spaces with $\gamma$ approximation ratio, $O(\log m)$ amortized recourse and $\text{poly}(d) \cdot (m+n)^{O(1/\gamma)}$ amortized update time, for every sufficiently large constant $\gamma \geq 1$. Our result is the first efficient dynamic algorithm for the *non-uniform* dynamic facility location problem in high-dimensional Euclidean spaces. It also provides a stronger recourse bound than the existing solutions.

## 1. Introduction

Facility location is a fundamental task in unsupervised machine learning and a central problem in clustering theory, underpinning applications such as data summarization and network design. Its connection to objectives like $k$-median clustering, via Lagrangian relaxation, highlights its importance in identifying latent data structures. As a result, the problem has been extensively studied due to its broad applicability and theoretical significance. In particular, several approximation algorithms are known for the problem (Guha & Khuller, 1999; Jain et al., 2003; Li, 2013).

While fundamental in static contexts, modern data analysis frequently involves dynamic datasets characterized by point insertions and deletions, necessitating algorithms that maintain high-quality clustering solutions efficiently over time. Therefore, facility location has also been extensively studied in various computational models as the streaming model (Czumaj et al., 2013; Indyk, 2004; Lammersen & Sohler, 2008), the online model (Fotakis, 2008; Meyerson, 2001) and dynamic algorithm model (Bhattacharya et al., 2024; 2022; Cohen-Addad et al., 2019; Cygan et al., 2018; Goranci et al., 2018; Guo et al., 2020) and many more. A particularly interesting model in this setting is the dynamic algorithm model for its ability to capture real-world scenarios where one is interested in ensuring solution stability, typically measured via recourse (the magnitude of changes per update), alongside approximation quality and computational efficiency. Addressing these competing requirements for evolving data in unsupervised learning has become a focus of recent research (Bateni et al., 2023; Bhattacharya et al., 2025a;b; 2023; Chan et al., 2018; Cohen-Addad et al., 2019; Cruciani et al., 2024; Fichtenberger et al., 2021; Forster & Skarlatos, 2025; Guo et al., 2021; Jaghargh et al., 2019; La Tour et al., 2024; Łącki et al., 2024; Lattanzi et al., 2020; Lattanzi & Vassilvitskii, 2017).

Although significant progress addresses dynamic facility location under uniform costs (Bhattacharya et al., 2024; 2022; Cohen-Addad et al., 2019) where efficient stable algorithms with good approximation are known, the general non-uniform cost setting, where facility opening costs vary, remains less explored despite its prevalence in real-world scenarios (e.g., differing intrinsic values or costs associated with potential centers). This paper makes significant progress in designing dynamic, efficient and stable algorithms in this setting. In particular, it investigates non-uniform dynamic facility location within high dimensional Euclidean spaces and shows that it is possible to obtain an efficient algorithm in this setting.

**Problem Definition.** Let $F$ and $D$ respectively denote the set of facilities and the set of clients in a $d$-dimensional Euclidean space[1]. Each facility $i \in F$ has an *opening cost* $f_i > 0$. Let $\text{dist}(i, j)$ denote the distance between a client

---

[1]Department of Computer Science, University of Warwick. [2]Google. Correspondence to: Nikos Parotsidis <nikosp@google.com>.

*Proceedings of the 43rd International Conference on Machine Learning*, Seoul, South Korea. PMLR 306, 2026. Copyright 2026 by the author(s).

---

[1]Our results are not restricted to the Euclidean setting and apply to any metric spaces admitting efficient dynamic Nearest Neighbor (NN) Oracle access. Although in the presentation we focus on Euclidean metric for sake of simplicity.

$j \in D$ and a facility $i \in F$. We consider a setting where the underlying input changes via a sequence of *updates*; each update inserts/deletes a client $j$ in $D$. Let $m = |F|$ and $n = |D|$ respectively denote the number of facilities and clients. The value of $m$ remains fixed since the set of facilities does not undergo any updates, whereas the value of $n$ changes over time. We define the *recourse* of our algorithm to be the number of changes in the assignments of clients to facilities, per update. Let $\Delta$ denote the aspect ratio of the underlying metric space. For simplicity, we assume that all the distances and the facility opening costs are polynomially bounded by $m^2$. Specifically, this means that $1/\text{poly}(m) \leq f_i \leq \text{poly}(m)$ for all $i \in F$; and $1/\text{poly}(m) \leq \text{dist}(i,j) \leq \text{poly}(m)$ for all $i \in F, j \in D$ at all times.

At any given point in time, a valid solution to the current input instance is defined by a subset of *open* facilities $F^\star \subseteq F$, and an assignment $\sigma : D \to F^\star$ of clients to the open facilities. This results in a total *facility-opening cost* of $\sum_{i \in F^\star} f_i$, and a total *connection cost* of $\sum_{j \in D} \text{dist}(\sigma(j), j)$. The objective is to minimize the sum of the total facility-opening cost and the total connection cost. For any $\alpha \geq 1$, we say that a given solution is $\alpha$-*approximate* iff its objective is at most $\alpha$ times the optimal objective.

In the dynamic setting, throughout the sequence of updates we wish to explicitly maintain an (approximately) optimal facility-location solution. The time taken by the algorithm to process an update is referred to its *update time*. Our algorithm incurs one unit of *client-recourse* whenever a client gets reassigned to a different facility in the maintained solution, and it incurs one unit of *facility-recourse* whenever it opens/closes a facility in the maintained solution. The *recourse* of the algorithm is defined to be the sum of its client-recourse and facility-recourse, per update. We wish to design an algorithm that has small approximation ratio, update time and recourse.

**Our Result.** We provide the first efficient algorithm for Dynamic Euclidean Non-Uniform Facility Location problem. Our algorithm gives both small update time, logarithmic recourse and constant approximation. More formally,

**Theorem 1.1.** *For every sufficiently large constant $\gamma \geq 1$, there is a randomized algorithm for dynamic facility location in $d$-dimensional Euclidean spaces with $\gamma$-approximation ratio, $O(\log m)$ amortized recourse and $\text{poly}(d) \cdot (m + n)^{O(1/\gamma)}$ amortized update time. The algorithm works against an adaptive adversary and handles each update correctly with high probability.*

**Remarks.** In high-dimensional Euclidean space, Theorem 1.1 presents almost the best trade-off between approx-

imation ratio and update time that one can achieve using the currently available tools. Indeed, even in the static sequential setting, the state-of-the-art algorithm for the high-dimensional Euclidean facility location problem has $\text{poly}(\gamma)$-approximation ratio and $O((m+n)^{1+1/\gamma})$ running time. So, improving our *dynamic* $O(m + n)^{O(1/\gamma)}$ update time bound would lead to an improvement over the best known static sequential algorithm for this problem, which will in turn require us to bypass the current best guarantees for locality sensitive hashing (LSH); this would be considered as a major breakthrough.

Furthermore, note that the $\text{poly}(d)$ dependence in the update time bound of Theorem 1.1 can be easily removed by invoking the Johnson-Lindenstrauss (JL) dimensionality reduction on the fly to reduce the dimension to $d = O(\log(m+n))$. However, we emphasize that if randomized JL is applied dynamically, the algorithm's internal updates depend on previous queries, meaning it no longer works against an adaptive adversary, but rather an oblivious one.

**Some Relevant Prior Work.** The literature on dynamic clustering is very vast, here we focus on the closest papers to the current work. The work of (Bhattacharya et al., 2024) considered a special case of the problem, where every facility has the *same opening cost*, and furthermore there is a facility at the location of each client. For every sufficiently large constant $\gamma \geq 1$, they obtained a $\gamma$-approximation with $\text{poly}(d) \cdot n^{O(1/\gamma)}$ update time and $O(1)$ amortized *facility recourse*. We note that this is a weaker notion of recourse from the one that we consider (facility *plus* client recourse). Specifically, facility recourse can be much smaller than client recourse, particularly when a facility serving a large number of clients is opened or closed.

The work of (Bhattacharya et al., 2022) considered the same setting as defined above (i.e., non-uniform facility opening costs, and our notion of recourse), but in general metric spaces. They obtained a $O(1)$-approximate dynamic algorithm with $O(\log m)$ recourse and $\tilde{O}(m)$ update time. Note that this $\Omega(m)$ factor in the update time is unavoidable in general metric spaces since any static algorithm that returns a non-trivial approximation in this setting has a running time of $\Omega(nm)$.[3]

### 1.1. Our Technical Contributions

Similar to (Bhattacharya et al., 2022), we dynamize the static-sequential greedy algorithm for facility location. This requires us to maintain a *hierarchical partition* where each client gets assigned to a *level* (see Section 2.1 for more details). Now, the algorithm by (Bhattacharya et al., 2022)

---

[2]Assuming distances and opening costs are polynomially bounded is a standard assumption in clustering literature, allowing us to replace aspect ratio terms with $O(\log m)$.

[3]In this paper we can bypass this limitation by leveraging the structure of the Euclidean space and by trading off approximation for running time.

needs to maintain an invariant (Invariant 3 in their paper), which says that if a client $j$ gets assigned to facility $i$ at level $k$, then we must have $k \geq k(i, j) :\simeq \log \text{dist}(i, j)$. In other words, if we fix a client $j$, then for every facility $i$ there is a minimum *threshold level* $k(i, j)$, such that $j$ must be at a level $\geq k(i, j)$ if it is to get assigned to $i$. We face an insurmountable hurdle if we wish to maintain this invariant in the Euclidean setting, for the following reason.

Essentially, for each level $k$, we need a "key data structure" to keep track of the (approximate) cheapest cluster that can be formed by moving some existing clients down from their current levels. In the algorithm of (Bhattacharya et al., 2022), this key data structure consists of $m$ subroutines, one for each facility. Each such subroutine (say) $S_{i,k}$, for level $k$ and facility $i$, keeps track of all clients $j$ with $k(i, j) \leq k$ that are at levels $> k$: These are precisely the clients which might form a cluster with facility $i$ at level $k$ leading to a violation of Invariant 2.4 and Invariant 2.5. Accordingly, whenever a client $j$ moves up or down one level in the algorithm of (Bhattacharya et al., 2022), we need to check each of the $m$ facilities to decide which of the subroutines $S_{i,k}$ need to update their set of relevant clients. In summary, the update time of such a scheme would inherently be $m$ times the client recourse, which is prohibitively expensive in the Euclidean setting.

In sharp contrast, our dynamic algorithm gets rid of this problematic invariant from (Bhattacharya et al., 2022) altogether. As a side effect, we also need to allow for the possibility of opening multiple clusters involving the same facility $i$, and accordingly we pay the opening cost of $i$ in more than one cluster. These crucial changes in the adaptation of the static sequential greedy algorithm, coupled with our novel data structures for *bi-chromatic closest pair oracle* and *cheapest cluster oracle* in high-dimensional Euclidean spaces (see Section 2.2) lead us to Theorem 1.1.

Finally, designing the data structures for bi-chromatic closest pair oracle and cheapest cluster oracle themselves requires new ideas, such as maintaining a maximal matching within a certain bipartite graph and introducing the concept of pseudo-average costs of a cluster; but due to space constraints we delegate these proofs to Section C.

## 2. Our Dynamic Algorithm

We use the term *cluster* to refer to an ordered pair $C = (i, A)$, where $i \in F$ is a facility and $A \subseteq D$ is a set of clients. We say that the facility $i$ is the *center* of the cluster $C = (i, A)$ and write $i = \text{center}(C)$. A cluster $(i, A)$ is of one of the following two types:

1. *Satellite* cluster. Here, the set $A$ consists of a single client (say) $j$ and the *cost* of the cluster equals $\text{cost}(i, A) := \text{dist}(i, j)$. We abuse the notation and

denote such a cluster by $(i, j)$.

2. *Regular* cluster. Here, the cost of the cluster is given by $\text{cost}(i, A) := f_i + \sum_{j \in A} \text{dist}(i, j)$.

We define the *average cost* of a cluster $C = (i, A)$ to be $\text{avg-cost}(C) := \frac{\text{cost}(C)}{|A|}$, and its *radius* to be $\text{radius}(C) := \max_{j \in A} \text{dist}(i, j)$. We maintain a *hierarchical partition* of clusters, defined by an ordered pair $(\mathcal{C}, \ell)$ and three fixed constant parameters $\alpha, \beta, \gamma \geq 1$. For reasons that will become clear later on, we further enforce that $\alpha \geq \beta + 9$; but $\alpha, \beta$ are independent of $\gamma$. This will be referred to as an $(\alpha, \beta, \gamma)$-*partition*. Here, $\mathcal{C}$ is a collection of clusters such that each client $j \in D$ appears in exactly one cluster in $\mathcal{C}$, and each cluster $C \in \mathcal{C}$ is assigned an integral *level* $\ell(C) \in \mathbb{Z}$. For a cluster $C = (i, A) \in \mathcal{C}$, we define the level of a client $j \in A$ to be $\ell(j) := \ell(C)$.[4] For every level $k \in \mathbb{Z}$, we define the sets $D_{>k} := \{j \in D : \ell(j) > k\}$ and $F^{\star}_{\leq k} := \{i \in F : \exists \text{ a regular cluster } C = (i, A) \in \mathcal{C} \text{ with } \ell(C) \leq k\}$.

### 2.1. Invariants and Intuition

We maintain an $(\alpha, \beta, \gamma)$-partition $(\mathcal{C}, \ell)$ that satisfy five invariants described below. Intuitively, these invariants are derived from the description of the greedy algorithm for (static) facility location (Jain et al., 2003). In each iteration, it picks a cluster $C = (i, A)$ with minimum average cost, defined over clients that are currently unassigned to any facility, and adds $C$ to its solution. The facility $i \in F$ is now open, and hence the algorithm resets the opening cost of facility $i$ to *zero* for all future operations. This gives rise to the satellite clusters; these are precisely those clusters whose centers have already been opened in previous iterations.

We can run the static greedy algorithm to build an $(\alpha, \beta, \gamma)$-partition. The clusters get assigned to the levels in a bottom-up manner (smallest levels first); since the average cost of the cluster being picked by the algorithm is monotonically non-decreasing across a series of consecutive iterations. This is formalized in Invariant 2.3. Next, Invariant 2.1 ensures that if we pick a satellite cluster with facility $i$ as center at level $k$, then the facility $i$ was open in some prior iteration of the greedy algorithm, by being part of a regular cluster at level $\leq k$. Invariant 2.4 and Invariant 2.5 together enforce the greedy property, namely, that in each iteration we pick a cluster with approximately minimum average cost. Finally, for reasons that will become clear during the proof of Theorem 2.8, we need Invariant 2.2, which upper bounds the radius of any cluster assigned to any given level.

**Invariant 2.1.** For every satellite cluster $C = (i, j) \in \mathcal{C}$, we have $i \in F^{\star}_{\leq \ell(C)}$.

---

[4]Since each client appears in exactly one cluster, the level assigned to a client is unique.

**Invariant 2.2.** For every cluster $C \in \mathcal{C}$, we have $\mathtt{radius}(C) \leq \beta \cdot 2^{\ell(C)}$.

**Invariant 2.3.** For every cluster $C \in \mathcal{C}$, we have $\mathtt{avg\text{-}cost}(C) \leq \alpha \cdot 2^{\ell(C)}$.

**Invariant 2.4.** For every level $k \in \mathbb{Z}$, client $j \in D_{>k}$ and facility $i \in F^{\star}_{\leq k}$, we have $\mathrm{dist}(i, j) \geq 2^k / \gamma$.

**Invariant 2.5.** For every level $k \in \mathbb{Z}$, subset of clients $A \subseteq D_{>k}$ and facility $i \in F$, we have

$$\frac{f_i + \sum_{j \in A} \mathrm{dist}(i, j)}{|A|} \geq \frac{2^k}{\gamma}.$$

**Observation 2.6.** Any $(\alpha, \beta, \gamma)$-partition $(\mathcal{C}, \ell)$ gives a valid solution to the facility location problem. The objective value of the concerned solution is at most $\sum_{C \in \mathcal{C}} \mathtt{cost}(C)$.

*Proof.* Let $F^{\star} := \{i \in F : i = \mathtt{center}(C) \text{ for some regular cluster } C \in \mathcal{C}\}$. We open the facilities in $F^{\star}$. Now, consider any client $j \in D$, and let $C = (i, A) \in \mathcal{C}$ be the unique cluster containing the client $j$ (i.e., $j \in A$). We assign the client $j$ to the facility $i$. Invariant 2.1 guarantees that $i \in F^{\star}$. Hence, this gives a valid facility-location solution on the input instance $(F, D)$.

Next, note that in the above solution, the connection cost of a client is part of the cost of the cluster the client belongs to. Furthermore, Invariant 2.1 ensures that for each $i \in F^{\star}$, the facility opening cost $f_i$ is also part of the cost of some regular cluster in $\mathcal{C}$ that has $i$ as its center. Accordingly, the objective value of the concerned solution is at most $\sum_{C \in \mathcal{C}} \mathtt{cost}(C)$. $\square$

**Observation 2.7.** In any $(\alpha, \beta, \gamma)$-partition $(\mathcal{C}, \ell)$ satisfying the five invariants described above, we have $\ell(C) \in [L, U]$ for all $C \in \mathcal{C}$, where $L, U$ are two integers with $L < U$ and $U - L = \Theta(\log m)$.

*Proof.* Since the distances and the facility-opening costs are bounded polynomially in $m$, we can choose two integral parameters $L < U$ with $U - L = \Theta(\log m)$ such that $2^L$ is a sufficiently small polynomial in $m$ and $2^U$ is a sufficiently large polynomial in $m$. At this point, no cluster needs to get assigned to a level that is either smaller than $L$ or larger than $U$. $\square$

From now on, we fix the parameters $L$ and $U$, and work with only those $(\alpha, \beta, \gamma)$-partitions where each level lies in the range $[L, U]$. The next theorem guarantees our approximation ratio. Due to space constraints, we delegate the proof of Theorem 2.8 to Section A.

**Theorem 2.8.** *If an $(\alpha, \beta, \gamma)$-partition $(\mathcal{C}, \ell)$ satisfies the five invariants described above, then $\sum_{C \in \mathcal{C}} \mathtt{cost}(C)$ is at most $(4\alpha\gamma^2 \cdot (\beta + 1) + 2\alpha\gamma)$ times the optimal facility location objective.*

## 2.2. Two Dynamic Data Structures in High Dimensional Euclidean Spaces

We now present two dynamic data structures in high dimensional Euclidean spaces that will be crucially used by our algorithm. The missing proofs from this section are delegated to Section C.

**Bi-chromatic Closest Pair Oracle:** We denote this oracle by $\mathtt{Closest\text{-}Pair}(F, D, \gamma)$. Here, $F$ is a set of facilities,[5] $D$ is a set of clients, and $\gamma \geq 1$ is a fixed parameter. We refer to $\gamma$ as the *approximation ratio* of the oracle. The oracle supports the following operations.

- UPDATE: Insert/delete a point in the set $F$ or $D$.

- QUERY: In response to a query, it returns a $\gamma$-approximate bi-chromatic closest pair $(p, q) \in F \times D$. Specifically, the pair $(p, q) \in F \times D$ returned by the oracle satisfies

$$\mathrm{dist}(p, q) \leq \gamma \cdot \min_{(p', q') \in F \times D} \{\mathrm{dist}(p', q')\}.$$

**Lemma 2.9.** For every sufficiently large absolute constant $\gamma > 1$, there exists an implementation of a bi-chromatic closest pair oracle with $\gamma$ approximation ratio, which handles queries and updates in $\mathrm{poly}(d) \cdot (m + n)^{O(1/\gamma^2)}$ time. The oracle works correctly whp and is robust against an adaptive adversary.

**Cheapest Cluster Oracle:** We denote this oracle by $\mathtt{Cheap\text{-}Cluster}(F, D, \gamma)$. Here, $F$ is a set of facilities, $D$ is a set of clients, and $\gamma$ is a fixed parameter. We refer to $\gamma$ as the *approximation ratio* of the oracle. Let $\mathcal{C}^{\star}$ denote the set of all regular clusters defined over $(F, D)$. The oracle supports the following operations.

- UPDATE: Insert/delete a point in the set $D$. The set $F$ remains unchanged over time.

- VALUE-QUERY: In response to such a query, it returns a value $\lambda$ such that

$$\min_{C' \in \mathcal{C}^{\star}} \mathtt{avg\text{-}cost}(C') \leq \lambda \leq \gamma \cdot \min_{C' \in \mathcal{C}^{\star}} \mathtt{avg\text{-}cost}(C').$$

- SOLUTION-QUERY: In response to such a query, it returns a regular cluster $C = (i, A)$, with $i \in F$ and $A \subseteq D$, such that

$$\mathtt{avg\text{-}cost}(C) \leq \gamma \cdot \min_{C' \in \mathcal{C}^{\star}} \{\mathtt{avg\text{-}cost}(C')\}.$$

---

[5] Note that we reuse the symbols $F$ and $D$ in Section 2.2 as local dummy variables representing generic inputs to the oracles, distinct from the global sets defined in the introduction.

**Lemma 2.10.** For every sufficiently large absolute constant $\gamma > 1$, there exists an implementation of a cheapest cluster oracle with $\gamma$ approximation ratio and $\text{poly}(d) \cdot (m + n)^{O(1/\gamma^2)}$ update time. The oracle answers a value-query in $\text{poly}(d) \cdot (m + n)^{O(1/\gamma^2)}$ time. Furthermore, it answers a solution-query in $\text{poly}(d) \cdot (m + n)^{O(1/\gamma^2)} \cdot |A|$ time, where $C = (i, A)$ is the regular cluster returned in response to the query. The oracle works correctly whp and is robust against an adaptive adversary.

**Handling Non-Uniform Opening Costs.** To handle non-uniform facility opening costs (as detailed in Appendix C.3), we partition the facilities into $\Theta(\log m)$ groups based on their opening costs, discretizing them in powers of 2. We scale the costs to be uniform within each group and maintain a separate cheapest cluster oracle for each group. Upon receiving a query, we query all $\Theta(\log m)$ oracles and return the cluster with the minimum average cost. This extends our results to the non-uniform setting, while increasing the update and query time by an $O(\log m)$ factor.

### 2.3. Handling an Update

In this section, we present our algorithm for handling an update (client insertion/deletion) in the dynamic setting. We start by describing some basic data structures that our algorithm will use.

**Basic Data Structures.** For every facility $i \in F$ and every level $k \in [L, U]$, define the sets $\mathcal{C}_i^{\star}(k) := \{C \in \mathcal{C} : i = \text{center}(C), \ell(C) = k, \text{ and } C \text{ is a regular cluster}\}$ and $\mathcal{C}_i(k) := \{C \in \mathcal{C} : i = \text{center}(C), \ell(C) = k, \text{ and } C \text{ is a satellite cluster}\}$.

We define $\mathcal{C}_i^{\star} := \bigcup_{k \in [L, U]} \mathcal{C}_i^{\star}(k)$, $\mathcal{C}_i := \bigcup_{k \in [L, U]} \mathcal{C}_i(k)$. Next, for every $i \in F$, we define $\ell(i) := \min\{k \in [L, U] : \mathcal{C}_i^{\star}(k) \neq \emptyset\}$ if $i$ is the center of at least one regular cluster (note that there can be multiple regular clusters with the exact same center) in $(\mathcal{C}, \ell)$, and $\ell(i) := \infty$ otherwise. Our algorithm explicitly maintains the sets $\{\mathcal{C}_i^{\star}(k), \mathcal{C}_i(k)\}_{i \in F, k \in [L, U]}$ as doubly linked lists. Each entry in the linked list for $\mathcal{C}_i^{\star}(k)$ (resp. $\mathcal{C}_i(k)$) corresponds to a cluster $C \in \mathcal{C}_i^{\star}(k)$ (resp. $C \in \mathcal{C}_i(k)$). Using these linked lists, we can compute the value of $\ell(i)$ in $O(U - L) = O(\log m)$ time. Further, for each level $k \in [L, U]$ (see Observation 2.7), we maintain the sets $F_{\leq k}^{\star}$ and $D_{>k}$, and run the two oracles $\text{Closest-Pair}(F_{\leq k}^{\star}, D_{>k}, \gamma)$ and $\text{Cheap-Cluster}(F, D_{>k}, \gamma)$.

For every level $k$, the algorithm checks if it is possible to add a cluster at level $k$, consisting of clients that are all currently at levels $> k$. If this is the case then we say that such a level $k$ is *dirty*, and we run a sub-routine to fix it. More formally, we say that a level $k \in [L, U]$ is *satellite-dirty* if a value-query to $\text{Closest-Pair}(F_{\leq k}^{\star}, D_{>k}, \gamma)$

returns a value $< 2^k$, and *regular-dirty* if a value-query to $\text{Cheap-Cluster}(F, D_{>k}, \gamma)$ returns a value $< 2^k$. We say that the level $k$ is *dirty* if it is either regular-dirty or satellite-dirty. Finally, for every cluster $C = (i, A) \in \mathcal{C}$, we keep track of its *size* (given by $|A|$) and cost (given by $\text{cost}(C)$). Whenever a client moves in or out of a cluster, its size and cost are updated in $O(1)$ time. Accordingly, we can calculate the average cost of a cluster (given by $\text{avg-cost}(C)$) also in $O(1)$ time.

The next two observations follow from the definitions of the two oracles in Section 2.2 and that of regular-dirty and satellite-dirty levels.

**Observation 2.11.** If a level $k$ is *not* regular-dirty, then Invariant 2.5 holds for level $k$. Otherwise, if the level $k$ *is* regular-dirty, then a solution-query to $\text{Cheap-Cluster}(F, D_{>k}, \gamma)$ returns a regular cluster $C = (i, A)$, with $i \in F$ and $A \subseteq D_{>k}$, such that $\text{avg-cost}(C) < 2^k$.

**Observation 2.12.** If a level $k$ is *not* satellite-dirty, then Invariant 2.4 holds for level $k$. Otherwise, if the level $k$ *is* satellite-dirty, then a solution-query to $\text{Closest-Pair}(F_{\leq k}^{\star}, D_{>k}, \gamma)$ returns a satellite cluster $C = (i, j) \in F_{\leq k}^{\star} \times D_{>k}$ such that $\text{avg-cost}(C) = \text{radius}(C) = d_{ij} < 2^k$.

**Handling the insertion of a client $j$.** We pick any facility $i \in F$, **create** a regular cluster $C = (i, \{j\})$ and assign it to level $U$. Thus, we set $\mathcal{C} \leftarrow \mathcal{C} \cup C$, $\ell(C) \leftarrow U$, and $\mathcal{C}_i^{\star}(U) \leftarrow \mathcal{C}_i^{\star}(U) \cup \{C\}$. At this point, we call the subroutine FIX-INVARIANTS(), as described in Algorithm 2.

**Handling the deletion of a client $j$.** Let $C = (i, A)$ be the unique cluster in $\mathcal{C}$ that the client $j$ was part of. We first call the subroutine REMOVE-CLIENT$(C, j)$, as described in Algorithm 1. Finally, we call the subroutine FIX-INVARIANTS(), as in Algorithm 2.

**The subroutine** REMOVE-CLIENT$(C, j)$**.** The subroutine takes as input a cluster $C = (i, A) \in \mathcal{C}$ at level $\ell(C) = k$ and a client $j \in A$, and removes $j$ from $C$. If $C$ is a satellite cluster, then it **destroys** $C$ after removing its only client $j$. Otherwise, if $C$ is a regular cluster then it simply sets $A \leftarrow A \setminus \{j\}$. An interesting case arises if we observe that $A = \emptyset$ at this point. Normally, we would like to **destroy** such an empty cluster. But since $C$ is regular, **destroying** it might lead to a violation of Invariant 2.1. This can occur only if $A = \emptyset$, $\mathcal{C}_i^{\star}(k) \setminus \{C\} = \emptyset$ and $\bigcup_{k'=k}^{U} \mathcal{C}_i(k') \neq \emptyset$. In this case, we pick a satellite cluster $C' = (i, j') \in \bigcup_{k'=k}^{U} \mathcal{C}_i(k')$ with minimum level, move the client $j'$ into $C$, move the cluster $C$ to level $\ell(C') \geq k$, and **destroy** $C'$. This step ensures that Invariant 2.1 continues to hold. Just before the call REMOVE-CLIENT$(C, j)$, the satellite cluster $C' = (i, j')$ satisfied Invariant 2.2, and so $\text{radius}(C') \leq 2^{\ell(C')}$. Thus, just after we move $j'$ into

**Algorithm 1:** REMOVE-CLIENT$(C, j)$

---

Let $C = (i, A)$ and $\ell(C) = k$
**if** *C is a satellite cluster* **then**
    $\mathcal{C}_i(k) \leftarrow \mathcal{C}_i(k) \setminus \{C\}$
    **destroy** the cluster $C$ and set $\mathcal{C} \leftarrow \mathcal{C} \setminus \{C\}$
**else**
    $A \leftarrow A \setminus \{j\}$
    **if** $A = \emptyset$ **then**
        **if** $\mathcal{C}_i^\star(k) \setminus \{C\} = \emptyset$ *and* $\bigcup_{k'=k}^U \mathcal{C}_i(k') \neq \emptyset$ **then**
            Let $k' \in [k, U]$ be the minimum level at
              which $\mathcal{C}_i(k') \neq \emptyset$
            Pick any satellite cluster
              $C' = (i, j') \in \mathcal{C}_i(k')$
            $A \leftarrow A \cup \{j'\}$ and $\ell(C) \leftarrow k'$
            $\mathcal{C}_i^\star(k) \leftarrow \mathcal{C}_i^\star(k) \setminus \{C\}$ and
              $\mathcal{C}_i^\star(k') \leftarrow \mathcal{C}_i^\star(k') \cup \{C\}$
            **destroy** the cluster $C'$ and set
              $\mathcal{C} \leftarrow \mathcal{C} \setminus \{C'\}$
        **else**
            **destroy** the cluster $C$ and set $\mathcal{C} \leftarrow \mathcal{C} \setminus \{C\}$
    **if** $A \neq \emptyset$ **then**
        **while** *C violates Invariant 2.3* **do**
            MOVE-UP$(C)$

---

$C$ we clearly have $\texttt{radius}(C) = \texttt{radius}(C') \leq 2^{\ell(C')}$. Finally, if $A \neq \emptyset$, removing a client might increase the average cost, potentially violating Invariant 2.3. To fix this, we repeatedly call MOVE-UP$(C)$ as long as Invariant 2.3 is violated. This extra step guarantees that all invariants are immediately restored after a deletion, without altering our asymptotic time and recourse bounds. This implies the following observation.

**Observation 2.13.** Just after a call to REMOVE-CLIENT$(C, j)$, Invariants 2.1 and 2.2 continue to hold.

**The subroutine** FIX-INVARIANTS$()$**.** This is described in Algorithm 2. At the start of any given iteration of the outer **while** loop, Invariant 2.1, Invariant 2.2 and Invariant 2.3 hold. However, Invariant 2.4 or Invariant 2.5 might be violated (in which case there exists some dirty level, as per Observation 2.11 and Observation 2.12). Accordingly, if there exists a dirty level $k \in [L, U]$, we identify a cluster $C = (i, A)$ that violates Invariant 2.4 or Invariant 2.5, by making appropriate queries to the two oracles from Section 2.2. However, we need to be careful about one specific issue. Suppose that we receive a regular cluster $C' = (i, A')$ in response to a solution-query to Cheap-Cluster$(F, D_{>k}, \gamma)$. It is easy to verify from the above pseudo-code and Observation 2.11 that in this case $\texttt{avg-cost}(C') < 2^k$. Nevertheless, the radius of this cluster might be larger than $\beta \cdot 2^k$, which would lead to a violation of Invariant 2.2. This is the reason why we call the subroutine PRUNE$(C', k)$ (see Algorithm 3) in this

situation. The subroutine PRUNE$(C', k)$ simply removes all the clients from $C'$ that are more than $\beta \cdot 2^k$ distance away from its center, and returns the resulting cluster $C$. Since initially we had $\texttt{avg-cost}(C') < 2^k$, the pruning step does not increase the average cost of the resulting cluster. So, we continue to have $\texttt{avg-cost}(C) < 2^k$. Also, by definition, at the end of the pruning step, we have $\texttt{radius}(C) \leq \beta \cdot 2^k$. Moreover, a simple averaging argument implies that at most $1/\beta$ fraction of the clients in $C'$ get pruned, for otherwise we would have $\texttt{avg-cost}(C') \geq 2^k$. This discussion leads to the following observations.

**Algorithm 2:** FIX-INVARIANTS$()$

---

**while** *there exists some dirty level $k \in [L, U]$* **do**
    **if** *level $k$ is satellite-dirty* **then**
        $C \leftarrow$ Closest-Pair$(F_{\leq k}^\star, D_{>k}, \gamma)$.
    **else**
        $C' \leftarrow$ Cheap-Cluster$(F, D_{>k}, \gamma)$
        $C \leftarrow$ PRUNE$(C', k)$
    CREATE-CLUSTER$(C, k)$
    **while** *there is some regular cluster $C'$ violating*
    *Invariant 2.3* **do**
        MOVE-UP$(C')$

---

**Observation 2.14.** Within FIX-INVARIANTS$()$, a call to PRUNE$(C' = (i, A'), k)$ takes $O(|A'|) = O(\beta \cdot |A|)$ time, where $C = (i, A)$ is the cluster returned by PRUNE$(C', k)$. We also have $\texttt{avg-cost}(C) < 2^k$ and $\texttt{radius}(C) \leq \beta \cdot 2^k$, and so $C$ satisfies Invariant 2.2 and Invariant 2.3 at level $k$. Finally, since $C$ is a regular cluster, assigning $C$ to level $k$ does not violate Invariant 2.1.

**Observation 2.15.** Within FIX-INVARIANTS$()$, just before the call to the subroutine CREATE-CLUSTER$(C, k)$, the cluster $C$ satisfies Invariant 2.2 and Invariant 2.3 at level $k$. Furthermore, assigning $C$ to level $k$ does not violate Invariant 2.1.

At this stage, the subroutine FIX-INVARIANTS$()$ makes a call to CREATE-CLUSTER$(C, k)$, as described in Algorithm 4. Let $C = (i, A)$. This subroutine creates the cluster $C$ at level $k$; and in the process *steals* the clients $j \in A$ from their existing clusters. Note that all the clients in $A$ strictly decrease their levels during this event. This leads to the following observation.

**Observation 2.16.** A call to CREATE-CLUSTER$(C = (i, A), k)$ takes $O(|A|)$ time and decreases the level of every client $j \in A$. At the end of the call to CREATE-CLUSTER$(C = (i, A), k)$, Invariant 2.1 and Invariant 2.2 continue to hold for all clusters, and Invariant 2.3 holds for the cluster $C$.

*Proof.* Follows from Observation 2.13, Observation 2.15, and the pseudo-code in Algorithm 4. $\qquad\square$

---

**Algorithm 3:** PRUNE$(C', k)$

---

Let $C' = (i, A')$
$A \leftarrow \emptyset$
**for** $j \in A'$ **do**
   **if** $d_{ij} \leq \beta \cdot 2^k$ **then**
      $A \leftarrow A \cup \{j\}$
**return** $C = (i, A)$

---

**Algorithm 4:** CREATE-CLUSTER$(C, k)$

---

Let $C = (i, A)$
**for** $j \in A$ **do**
   Let $C' = (i', A') \in \mathcal{C}$ be the unique cluster which
   contains the client $j$ (i.e., $j \in A'$)
   REMOVE-CLIENT$(C', j)$
**create** the cluster $C$, and set $\mathcal{C} \leftarrow \mathcal{C} \cup \{C\}$ and
$\ell(C) \leftarrow k$
**if** $C$ *is a satellite cluster* **then**
   $\mathcal{C}_i(k) \leftarrow \mathcal{C}_i(k) \cup \{C\}$
**else**
   $\mathcal{C}_i^\star(k) \leftarrow \mathcal{C}_i^\star(k) \cup \{C\}$

---

At the end of the call to CREATE-CLUSTER$(C, k)$, some other regular clusters $C'$ might violate Invariant 2.3; these are precisely those clusters that lose some clients who subsequently join the newly created cluster $C$. Note that we don't need to worry about a satellite cluster violating Invariant 2.3, because those satellite clusters that lose a client get destroyed (see the pseudo-code in Algorithm 1). Thus, we run the inner **while** loop in Algorithm 2. In each iteration of this inner **while** loop, we identify a regular cluster $C'$ violating Invariant 2.3, and call the subroutine MOVE-UP$(C')$, as described in Algorithm 5.

Ideally, a call to MOVE-UP$(C)$ should move the cluster $C$ up by one level, since it violates Invariant 2.3. However, we need to be careful about the following issue. Let $C = (i, A)$ and $\ell(C) = k$. It might be the case that there are other satellite clusters, with facility $i$ as their center, at level $k$. If we move the regular cluster $C$ to level $k + 1$, then all these satellite clusters might now violate Invariant 2.1. To address this concern, we move the clients from all these satellite clusters into the cluster $C$, destroy the satellite clusters, and after this move up the cluster $C$ by one level if it still continues to violate Invariant 2.1. For every satellite cluster $C' = (i, j')$ that get destroyed during this process, we had $\text{dist}(i, j) \leq \beta \cdot 2^k$ (see Invariant 2.2). Accordingly, at the end of the subroutine, the cluster $C$ continues to satisfy Invariant 2.2. This leads to the following observation.

**Observation 2.17.** A call to the subroutine MOVE-UP$(C = (i, A))$, with $\ell(C) = k$, takes $O(|\mathcal{C}_i(k)|)$ time; i.e., the time complexity is proportional to the number of satellite clusters

with $i$ as the center at level $\ell(C)$. Throughout the duration of the call, Invariant 2.1 and Invariant 2.2 continue to hold.

---

**Algorithm 5:** MOVE-UP$(C)$

---

Let $C = (i, A)$ and $\ell(C) = k$
**for** $C' \in \mathcal{C}_i(k)$ **do**
   Let $C' = (i, j')$
   $A \leftarrow A \cup \{j'\}$
   $\mathcal{C}_i(k) \leftarrow \mathcal{C}_i(k) \setminus \{C'\}$
   **destroy** the cluster $C'$ and set $\mathcal{C} \leftarrow \mathcal{C} \setminus \{C'\}$
**if** $avg\text{-}cost(C) > \alpha \cdot 2^k$ **then**
   $\ell(C) \leftarrow k + 1$
   $\mathcal{C}_i^\star(k) \leftarrow \mathcal{C}_i^\star(k) \setminus \{C\}$
   $\mathcal{C}_i^\star(k + 1) \leftarrow \mathcal{C}_i^\star(k + 1) \setminus \{C\}$

---

The correctness of our dynamic algorithm immediately follows from the above discussion.

### 2.4. Analysis: Proof of Theorem 1.1

We first define the notion of *movement-cost*. Our algorithm incurs one unit of movement-cost each time one of the following events occur: (1) A client increases its level by one. This contributes one unit of *up-cost*. (2) A client decreases its level by one. This contributes one unit of *down-cost*. (3) A client $j$ moves from being part of a satellite cluster $(i, j)$ to being part of a regular cluster $(i, A)$, without changing the facility $i$ as its center. For example, this might occur within the subroutines MOVE-UP$(C)$ and and REMOVE-CLIENT$(C, j)$. This contributes one unit of *switch-cost*.

The total movement-cost is the sum of the total up-costs, total down-costs and the total switch-costs.

**Corollary 2.18.** The total recourse of our algorithm is at most a constant times its total movement-cost plus the total number of updates it has to process.

*Proof.* We incur a *client-recourse* of one whenever a client gets reassigned to a different facility. In our algorithm, such an event occurs only during a call to CREATE-CLUSTER$(C, k)$, and during that call the concerned client also moves down to a lower level (see Observation 2.16). Accordingly, the total client-recourse of our algorithm is upper bounded by its total down-cost.

Furthermore, in our algorithm a facility $i \in F$ switches from being open to closed only when a regular cluster with $i$ as its center gets destroyed. This can happen only during a call to REMOVE-CLIENT$(C, j)$; see the last line in the pseudo-code of Algorithm 1. However, for such an event to occur, first a client $j$ needs to get removed from the concerned cluster $C$. This client $j$ either gets deleted from the input altogether, or moves down to a lower level to a newly created cluster (see the pseudo-codes in Algorithm 2

and Algorithm 4, and Observation 2.16). In other words, each unit of facility-recourse can be charged to either some unit of down-cost or some specific update.  □

In Lemma 2.19, we bound the amortized movement-cost of our algorithm. Subsequently, Lemma 2.20 shows that our amortized update time is at most the amortized movement-cost, with a multiplicative $\text{poly}(d) \cdot (m + n)^{O(1/\gamma)}$ factor overhead (using the two oracles from Section 2.2). Theorem 1.1 now follows from Observation 2.6, Theorem 2.8, Corollary 2.18, Lemma 2.19 and Lemma 2.20.

**Lemma 2.19.** Our dynamic algorithm has an amortized movement-cost of $O(\log m)$.

**Lemma 2.20.** Our dynamic algorithm has an amortized update time of $\text{poly}(d) \cdot (m + n)^{O(1/\gamma^2)}$.

**Remark on bounds.** Note that while Theorem 2.8 establishes an approximation ratio of $O(\gamma^2)$ and Lemma 2.20 bounds the update time by $\text{poly}(d) \cdot (m + n)^{O(1/\gamma^2)}$, we can seamlessly bridge this gap. By substituting $\gamma$ for $\gamma^2$ in our parameterization, we yield the $O(\gamma)$ approximation ratio and the $\text{poly}(d) \cdot (m + n)^{O(1/\gamma)}$ update time exponent stated in Theorem 1.1.

Due to space constraints, we delegate the proof of Lemma 2.20 to Section B. We devote the rest of this section to the proof of Lemma 2.19.

We use a *token-based* argument. Each client $j \in D$ stores 3 units of token at each level $k \in [L, \ell(j)]$. Also, each regular cluster in $\mathcal{C}$ has a *bank-account* associated with it, where it stores some nonnegative number of tokens, which can vary over time. In contrast, each satellite cluster in $\mathcal{C}$ also has a *bank-account* associated with it, which always stores 1 unit of token. When a client gets inserted, it gets assigned to a newly created regular cluster $C$ at level $U$. At initialization, the bank account for $C$ gets zero tokens. Similarly, if our algorithm creates a regular cluster while handling an update, at initialization the bank-account associated with the regular cluster gets zero tokens. At initialization, the client $j$ creates 3 tokens for each level $k \in [L, U]$ that will be released during the execution of the algorithm. In contrast, if a client $j$ that currently belongs to a regular cluster $C = (i, A) \in \mathcal{C}$ gets deleted, then we deposit 1 unit of token to the bank-account associated with $C$. We will now show that these tokens are sufficient to pay for the total movement-cost, which will imply Lemma 2.19.

**Accounting for switch-costs:** Suppose that a client $j$ moves from being part of a satellite cluster $C' = (i, j)$ to a regular cluster $C = (i, A)$. This might happen during a call to REMOVE-CLIENT$(C, j)$, CREATE-CLUSTER$(C, k)$ or MOVE-UP$(C)$. Then, cluster $C'$ gets destroyed, and the 1 unit of token associated with its bank-account is used to pay for the switch-cost we incur. On the other hand, a satellite cluster can get created only during a call to CREATE-CLUSTER$(C, k)$, and this leads to the client decreasing its level (see Observation 2.16). Then, we create a bank-account for this new satellite cluster and put 1 unit of token into it. We account for this while discussing down-costs below.

**Accounting for down-costs:** We incur down-costs only during calls to CREATE-CLUSTER$(C, k)$. Specifically, this happen when a client $j$, which was part of a cluster $C' = (i', A')$ at level $\ell(C') = k' > k$, leaves the cluster $C'$ and moves down to level $k$ to join the newly created cluster $C$. Due to this event, the number of tokens associated with client $j$ decreases by $3(k' - k)$; we say that the client $j$ *releases* $3(k' - k)$ tokens. We split up these tokens into three parts: $T_1 = T_2 = T_3 = (k' - k)$. We use $T_1$ tokens to pay for the down-cost we incur as the client moves down $k' - k$ levels. If $C'$ were a regular cluster, then we deposit $1 \leq T_2$ token into the bank account associated with $C'$. Finally, if $C$ is a satellite cluster, then we deposit $1 \leq T_3$ token into the bank-account associated with this new created satellite cluster (as dictated by our token based scheme). Next, consider the case where a client $j$ gets deleted. Just before this deletion, suppose that $j$ was part of some cluster $C$ at level $\ell(C) = k$. Then the client $j$ releases $3(k - L + 1)$ tokens at the time of this deletion. If $C$ were a regular cluster, then we add $1 \leq 3(k - L + 1)$ to the bank-account associated with $C$. This leads us to the following observation.

**Observation 2.21.** Whenever a regular cluster loses a client, it gains one token in its bank account.

**Accounting for up-costs:** First, we observe that a client moves up a level only when it is part of a regular cluster. Keeping this observation in mind, we start with two claims.

**Claim 2.22.** Consider the scenario where a regular cluster $C = (i, A)$ is moving up from level $k$ to level $k + 1$. Then, at this point in time, we have $|A| \leq f_i / ((\alpha - \beta) \cdot 2^k)$.

*Proof.* The cluster is moving up because it violates Invariant 2.3. Furthermore, Invariant 2.2 continues to hold throughout this event (see Observation 2.17). Hence, we get $\text{dist}(i, j) \leq \beta \cdot 2^k$ for all $j \in A$. Thus, it follows that $\frac{f_i + \beta \cdot 2^k \cdot |A|}{|A|} \geq \text{avg-cost}(C) > \alpha \cdot 2^k$. Rearranging the terms, we get $f_i > (\alpha - \beta) \cdot 2^k \cdot |A|$. The claim follows.  □

**Claim 2.23.** Consider the scenario where a regular cluster $C = (i, A)$ is created at level $k$. Then, at this point in time, we have $|A| \geq f_i / 2^k$.

*Proof.* When the regular cluster $C = (i, A)$ gets created at level $k$, by Observation 2.14 we have $\frac{f_i}{|A|} \leq \text{avg-cost}(C) < 2^k$. This implies that $|A| > f_i / 2^k$.  □

Now, let us track a regular cluster $C = (i, A)$ through its lifetime. Suppose that the cluster gets created at level $k$. Our algorithm ensures that the same cluster never decreases its level. (It can only get destroyed at some future time-step, after potentially moving up a sequence of levels.) Furthermore, whenever the cluster is moving up from level (say) $k'$ to $k' + 1$, for some $k' \geq k$, Claim 2.22 guarantees that $|A| \leq f_i/((\alpha - \beta) \cdot 2^{k'})$. During this step, we need to *replenish* $3 \cdot |A|$ tokens, because our scheme dictates that every client in $A$ now gets 3 additional tokens for level $k' + 1$. Furthermore, we incur an up-cost of $|A|$. Thus, during this step, the sum of the up-costs incurred and the tokens we need to replenish is $4 \cdot |A| \leq 4 \cdot f_i/((\alpha - \beta) \cdot 2^{k'})$. Summing over all levels $k' \geq k$, the total number of tokens we need to replenish plus the total up-cost we need to pay for the cluster $C$, throughout its entire lifetime, is at most $\sum_{k' \in [k, U]} \frac{4 f_i}{(\alpha - \beta) \cdot 2^{k'}} \leq \frac{8}{(\alpha - \beta)} \cdot \frac{f_i}{2^k}$. When the regular cluster $C$ was first created at level $k$, we had $|A| \geq f_i/2^k$ as per Claim 2.23. Subsequently, when the cluster moved up from level $k$ to level $k + 1$, we had $|A| \leq f_i/((\alpha - \beta) \cdot 2^k)$, as per Claim 2.23. Thus, in between these two time-steps, the cluster $C$ must have lost at least $\frac{f_i}{2^k} \cdot \left(1 - \frac{1}{(\alpha - \beta)}\right) = \frac{f_i}{2^k} \cdot \left(\frac{\alpha - \beta - 1}{\alpha - \beta}\right) \geq \frac{f_i}{2^k} \cdot \frac{8}{(\alpha - \beta)}$ many clients (since $\alpha \geq \beta + 9$). From Observation 2.21, we infer that just before the cluster $C$ makes its very first move (going from level $k$ to level $k+1$), it has at least $\frac{f_i}{2^k} \cdot \left(\frac{8}{\alpha - \beta}\right)$ tokens in its bank-account. This is sufficient to pay for the total up-cost, and replenish all the tokens for relevant clients, during the entire lifetime of the cluster $C$. This concludes the proof of Lemma 2.19.

## 3. Discussion and Open Problems

In this paper, we presented the first efficient dynamic algorithm for non-uniform facility location in high-dimensional Euclidean spaces. Specifically, we presented a randomized algorithm for dynamic facility location in $d$-dimensional Euclidean spaces with $\gamma$ approximation ratio, $O(\log m)$ amortized recourse and $\text{poly}(d) \cdot (m+n)^{O(1/\gamma)}$ amortized update time, for every sufficiently large constant $\gamma \geq 1$.

We now discuss two notes on practical applications of our algorithm. First, our theoretical algorithm allows for opening multiple clusters at the exact same facility location. This is strictly an *artifact of the analysis*; a practical implementation can simply open a physical facility the first time a theoretical cluster uses it, and close it when the last cluster at that location is destroyed. Second, the requirement for a large constant $\gamma$ in our analysis is an artifact of relying on theoretical Approximate Nearest Neighbor Search (ANNS) data structures (like LSH). Combining our framework with state-of-the-art, practical graph-based ANNS indices could yield a highly efficient method without large approximation

ratios or super-exponential time dependencies.

Finally, this work opens exciting directions for future research. Two prominent open problems are: (1) extending this dynamic framework to capacitated or prize-collecting/robust settings, and (2) proving a theoretical lower bound demonstrating whether an update time of $o((m + n)^{1/\gamma})$ inherently requires an $\Omega(\log m)$ recourse for non-uniform costs in general metric spaces.

## Acknowledgments

Sayan Bhattacharya is funded by the European Union (ERC grant, DYNALP, 101170133). Views and opinions expressed are however those of the author(s) only and do not necessarily reflect those of the European Union or the European Research Council Executive Agency. Neither the European Union nor the granting authority can be held responsible for them.

## Impact Statement

This paper presents work whose goal is to advance the field of Machine Learning. The nature of our work is mainly theoretical, and thus our insights might be used in various areas.

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

## A. Analysis of Approximation Ratio: Proof of Theorem 2.8

The proof will be via dual fitting. We first recall the LP-relaxation for this problem.

$$\text{Minimize} \qquad \sum_{i \in F} f_i \cdot y_i + \sum_{i \in F, j \in D} \text{dist}(i,j) \cdot x_{ij} \tag{1}$$

$$\text{s.t.} \sum_{i \in F} x_{ij} \;\geq\; 1 \quad \text{for all } j \in D. \tag{2}$$

$$x_{ij} \;\leq\; y_i \quad \text{for all } i \in F, j \in D. \tag{3}$$

$$x_{ij}, y_i \;\geq\; 0 \quad \text{for all } i \in F, j \in D. \tag{4}$$

The dual LP is as follows.

$$\text{Maximize} \qquad \sum_{j \in D} v_j \tag{5}$$

$$\text{s.t.} \sum_{j \in C} u_{ij} \;\leq\; f_i \text{ for all } i \in F. \tag{6}$$

$$v_j - u_{ij} \;\leq\; \text{dist}(i,j) \text{ for all } i \in F, j \in D. \tag{7}$$

$$v_j, u_{ij} \;\geq\; 0 \text{ for all } i \in F, j \in D. \tag{8}$$

**Remark.** W.l.o.g., in the dual we can assume that $u_{ij} := \max(0, v_j - \text{dist}(i,j))$ for all $i \in F, j \in D$.

**Setting the Dual Values.** Throughout the rest of this section, fix any $(\alpha, \beta, \gamma)$-partition $(\mathcal{C}, \ell)$ which satisfies the five invariants stated in Section 2. Based on $(\mathcal{C}, \ell)$, we set the dual variables as follows. We define

$$v_j \;:=\; 2^{\ell(j)} \qquad\qquad\qquad \text{for all } j \in D; \tag{9}$$

$$\hat{v}_j \;:=\; \frac{v_j}{4\gamma^2 \cdot (\beta + 1) + 2\gamma} \qquad \text{for all } j \in D; \tag{10}$$

$$\hat{u}_{ij} \;:=\; \max(0, \hat{v}_j - \text{dist}(i,j)) \qquad \text{for all } i \in F, j \in D. \tag{11}$$

Theorem 2.8 follows from Observation 2.6, Lemma A.1 and Lemma A.2, via dual fitting.

**Lemma A.1.** We have $\sum_{C \in \mathcal{C}} \text{cost}(C) \leq (4\alpha\gamma^2 \cdot (\beta + 1) + 2\alpha\gamma) \cdot \sum_{j \in D} \hat{v}_j$.

*Proof.* Consider any cluster $C = (i, A) \in \mathcal{C}$ and any client $j \in A$. By Invariant 2.3, we have $\text{cost}(C)/|A| \leq \alpha \cdot 2^{\ell(C)} = \alpha \cdot v_j$. Summing over all $j \in A$, we get $\text{cost}(C) \leq \alpha \cdot \sum_{j \in A} v_j$. Next, summing over all clusters $C \in \mathcal{C}$, we get

$$\sum_{C \in \mathcal{C}} \text{cost}(C) \leq \alpha \cdot \sum_{j \in D} v_j = \alpha \cdot (4\gamma^2 \cdot (\beta + 1) + 2\gamma) \cdot \sum_{j \in D} \hat{v}_j.$$

This concludes the proof. $\square$

**Lemma A.2.** The values $\{\hat{v}_j, \hat{u}_{ij}\}$ form a feasible solution to the dual LP.

We devote the rest of this section to the proof of Lemma A.2.

**Corollary A.3.** Consider any client $j \in D$ that gets assigned to a facility $i \in F$ in the $(\alpha, \beta, \gamma)$-partition $(\mathcal{C}, \ell)$.[6] Then, $\text{dist}(i,j) \leq \beta \cdot v_j$.

*Proof.* Let $C = (i, A) \in \mathcal{C}$ be the unique cluster which contains $j$ (i.e., $j \in A$). Accordingly, by Invariant 2.2, we have $\text{dist}(i,j) \leq \text{radius}(C) \leq \beta \cdot 2^{\ell(C)} = \beta \cdot 2^{\ell(j)} = \beta \cdot v_j$. $\square$

---

[6]Specifically, this means that there exists a cluster $C = (i, A) \in \mathcal{C}$ with $j \in A$.

**Lemma A.4.** Consider any facility $i \in F$ and any two clients $j, j_1 \in D$ such that $\mathrm{dist}(i,j) \leq v_j$ and $\mathrm{dist}(i,j_1) \leq v_{j_1}$. Then, $v_j \leq 2\gamma \cdot (\beta + 1) \cdot v_{j_1} + 2\gamma \cdot \mathrm{dist}(i,j)$.

*Proof.* If $v_j \leq v_{j_1}$, then the lemma is trivially true. So, henceforth, we assume that $v_{j_1} < v_j$. Since $v_{j_1} = 2^{\ell(j_1)}$ and $v_j = 2^{\ell(j)}$, this means that $\ell(j_1) \leq \ell(j) - 1$.

Let $C_1 = (i_1, A_1) \in \mathcal{C}$ be the unique cluster $j_1$ is assigned to, i.e., $j_1 \in A_1$. If $C_1$ is a regular cluster, then $i_1 \in F^{\star}_{\leq \ell(j_1)}$ by definition. Otherwise, $C_1$ is a satellite cluster, and on this occasion Invariant 2.1 guarantees that $i_1 \in F^{\star}_{\leq \ell(j_1)}$. As $\ell(j_1) \leq \ell(j) - 1$, Invariant 2.4 implies that

$$\mathrm{dist}(i_1, j) \geq \frac{2^{\ell(j)-1}}{\gamma} = \frac{v_j}{2\gamma}. \tag{12}$$

Furthermore, Corollary A.3 implies that

$$\mathrm{dist}(i_1, j_1) \leq \beta \cdot v_{j_1}. \tag{13}$$

From Equation (12), Equation (13) and triangle inequality, we derive that:

$$v_j \leq 2\gamma \cdot \mathrm{dist}(i_1, j) \leq 2\gamma \cdot (\mathrm{dist}(i_1, j_1) + \mathrm{dist}(i, j_1) + \mathrm{dist}(i, j)) \leq 2\gamma \cdot (\beta \cdot v_{j_1} + v_{j_1} + \mathrm{dist}(i,j)).$$

This concludes the proof. $\qquad \square$

**Proof of Lemma A.2.** Fix any facility $i \in F$, and consider the set of clients

$$C_i := \{j \in C : \hat{v}_j > \mathrm{dist}(i,j)\} = \{j \in C : \hat{u}_{ij} > 0\}.$$

We will show that $\sum_{j \in C_i} \hat{u}_{ij} \leq f_i$. Towards this end, let $C_i = \{j_1, \ldots, j_q\}$ with $v_{j_1} \leq v_{j_2} \leq \cdots \leq v_{j_q}$. Thus, Equation (9) gives us

$$\ell(j_1) \leq \ell(j_2) \leq \cdots \leq \ell(j_2). \tag{14}$$

Accordingly, setting $k = \ell(j_1) - 1$ and $A = C_i$ in Invariant 2.5, we get

$$\frac{1}{2\gamma} \cdot v_{j_1} = \frac{1}{2\gamma} \cdot 2^{\ell(j_1)} = \frac{2^k}{\gamma} \leq \frac{f_i + \sum_{j \in C_i} \mathrm{dist}(i,j)}{q}.$$

Rearranging the terms in the above inequality, we get

$$q \cdot v_{j_1} \leq 2\gamma \cdot f_i + 2\gamma \cdot \sum_{j \in C_i} \mathrm{dist}(i,j). \tag{15}$$

Now, from Lemma A.4 and Equation (15), we infer that

$$
\begin{aligned}
\sum_{j \in C_i} v_j &\leq 2\gamma \cdot (\beta + 1) \cdot q \cdot v_{j_1} + \sum_{j \in C_i} 2\gamma \cdot \mathrm{dist}(i,j) \\
&\leq 4\gamma^2 \cdot (\beta + 1) \cdot f_i + \left(4\gamma^2 \cdot (\beta + 1) + 2\gamma\right) \cdot \sum_{j \in C_i} \mathrm{dist}(i,j) \\
&\leq \left(4\gamma^2 \cdot (\beta + 1) + 2\gamma\right) \cdot \left(f_i + \sum_{j \in C_i} \mathrm{dist}(i,j)\right).
\end{aligned}
$$

Diving both sides by $\left(4\gamma^2 \cdot (\beta + 1) + 2\gamma\right)$, and rearranging the terms, we get:

$$\sum_{j \in C_i} \hat{u}_{ij} = \sum_{j \in C_i} (\hat{v}_j - \mathrm{dist}(i,j)) \leq f_i.$$

Finally, since $\hat{u}_{ij} := \max(0, \hat{v}_j - \mathrm{dist}(i,j))$, the dual constraint $v_j - u_{ij} \leq \mathrm{dist}(i,j)$ trivially holds for all $i \in F, j \in D$. This shows that $\{\hat{v}_j, \hat{u}_{ij}\}$ is a feasible solution to the dual LP. $\qquad \square$

## B. Bounding the Amortized Update Time: Proof of Lemma 2.20

Essentially, we will show that the update time of our algorithm is dominated by its movement-cost, up to a factor of $\text{poly}(d) \cdot (m + n)^{O(1/\gamma^2)}$. Towards this end, we consider each of the relevant subroutines used by our algorithm, one after another.

**The subroutine** REMOVE-CLIENT$(C, j)$. Using the relevant supporting data structures, each call to this subroutine can be implemented in $\tilde{O}(1)$ time.

**The subroutine** PRUNE$(C', k)$. By Observation 2.14, the time taken to implement a call to this subroutine is proportional (up to a $\beta$ factor, which is a constant) to the size of the cluster $C = (i, A)$ returned by it. This time complexity gets subsumed by the time complexity of the subsequent call to CREATE-CLUSTER$(C, k)$.

**The subroutine** CREATE-CLUSTER$(C, k)$. By Observation 2.16, the time taken to implement a call to this subroutine is proportional to the number of clients that decrease their levels due to this call. Hence, Lemma 2.19 implies that our algorithm spends an amortized $O(\log m)$ time per update to execute this subroutine.

**The subroutine** MOVE-UP$(C)$. By Observation 2.17, the time spent on a call to this subroutine is proportional to the number of clients that move from a satellite cluster with $i$ as the center to the cluster $C$. Thus, the time-complexity of this subroutine is subsumed by the switch-cost of our algorithm, and so Lemma 2.19 implies that we spend $O(\log m)$ amortized time per update to execute this subroutine.

**The subroutine** FIX-INVARIANTS$()$. From the pseudo-code in Algorithm 2, it follows that each call to CREATE-CLUSTER$(C = (i, A), k)$ is preceded by either a a solution-query to the oracle Closest-Pair$(F^\star_{\leq k}, D_{>k}, \gamma)$, or a solution-query to the oracle Cheap-Cluster$(F, D_{>k}, \gamma)$. In the former case, a solution-query to Closest-Pair$(F^\star_{\leq k}, D_{>k}, \gamma)$ takes $\text{poly}(d) \cdot (m + n)^{O(1/\gamma^2)}$ time, as per Lemma 2.9.

In the latter case, let $C' = (i, A')$ be the cluster returned by Cheap-Cluster$(F, D_{>k}, \gamma)$ in response to the solution-query. As per Lemma 2.10, this takes $|A'| \cdot \text{poly}(d) \cdot (m + n)^{O(1/\gamma^2)}$ time. By Observation 2.14, we have $|A'| = O(\beta \cdot |A|)$. Observation 2.16 guarantees that the time complexity of the subsequent call to CREATE-CLUSTER$(C, k)$ is $O(|A|)$. To summarize, the time spent on the solution-queries to Cheap-Cluster$(F, D_{>k}, k)$ is within a $\text{poly}(d) \cdot (m + n)^{O(1/\gamma^2)}$ factor of the time spent on the calls to CREATE-CLUSTER$(C, k)$, and we have previously argued that the amortized time complexity of the calls to CREATE-CLUSTER$(C, k)$ is $O(\log m)$. Thus, the amortized complexity of the time spent of the solution-queries to Cheap-Cluster$(F, D_{>k}, \gamma)$ is $\text{poly}(d) \cdot (m + n)^{O(1/\gamma^2)}$.

Finally, an iteration of the outer **while** loop in Algorithm 2 necessitates a value-query to each of the oracles Closest-Pair$(F^\star_{\leq k}, D_{>k}, \gamma)$ and Cheap-Cluster$(F, D_{>k}, \gamma)$, for each level $k \in [L, U]$, to determine whether or not there exists a dirty level. By Lemma 2.9 and Lemma 2.10, these value-queries take $\text{poly}(d) \cdot (m + n)^{O(1/\gamma^2)}$ time. For all but the last iteration of this outer **while** loop, these value-queries are followed by a call to CREATE-CLUSTER$(C, k)$. Using the same argument as in the preceding paragraph, we conclude that the amortized time complexity of these value-queries is also $\text{poly}(d) \cdot (m + n)^{O(1/\gamma^2)}$.

This concludes the proof of Lemma 2.20.

## C. Missing Proofs from Section 2.2

### C.1. Preliminaries

**Dynamic Nearest-Neighbor Oracle:** This is a dynamic data structure that maintains a point-set $S$ in a $d$-dimensional Euclidean space, and supports the following operations. (i) UPDATE: Insert/delete a point in $S$. (ii) QUERY: Given a point $q$ (not necessarily in $S$), return a $\gamma$-approximate nearest-neighbor $p_q \in S$ of $q$ within the set $S$, for some fixed parameter $\gamma \geq 1$. Specifically, the returned point $p_q \in S$ satisfies the condition that $\text{dist}(q, p_q) \leq \gamma \cdot \text{dist}(q, S)$, where $\text{dist}(q, S) := \min_{p \in S} \text{dist}(q, p)$. We refer to the parameter $\gamma$ as the approximation ratio of the oracle.

While efficient dynamic nearest-neighbors oracles have been previously known, an oracle that *works against an adaptive adversary* was first demonstrated very recently (Bateni et al., 2024). This property implies that the data structure can handle updates, which are dependent on the return values of the previous queries. This is crucial for the analysis of our algorithm, where this feedback loop is indeed present.

**Theorem C.1** ((Bateni et al., 2024))**.** *For every sufficiently large constant $\gamma \geq 1$, there exists a randomized dynamic nearest-neighbor oracle with $\gamma$-approximation ratio, which can handle updates and queries in $poly(d) \cdot n^{O(1/\gamma^2)}$ time where $n = |S|$ denotes the size of the input point-set. The data structure works correctly with high probability and is robust against an adaptive adversary.*

We note that the theorem formulation given in (Bateni et al., 2024) differs a little bit from the above one. First, the approximation ratio is $O(\gamma)$, and second there is an additional additive approximation equal to the minimum possible distance in the metric space. We note that we can achieve the result as given above by scaling $\gamma$ by a constant factor. Note that the query and update time remain $poly(d) \cdot n^{O(1/\gamma^2)}$.

Using Theorem C.1, it is known how to design a fast dynamic data structure which (approximately) keeps track of the sizes of the balls of a certain radius, around all the points within a dynamic point-set (Bhattacharya et al., 2024). This is summarized in the theorem below.

**Theorem C.2** ((Bhattacharya et al., 2024))**.** *For all $\lambda > 0$, $\kappa \geq 1$, there is a data structure $\mathcal{D}_{\lambda,\kappa}(S)$ for handling a dynamic point-set $S$. Specifically, it maintains a bit $b_{\lambda,\kappa}(p, S) \in \{0, 1\}$ for each point $p \in S$, and after each update (point insertion/deletion) in $S$, it reports all the points that change their $b_{\lambda,\kappa}(p, S)$ values. At every time-step, the following properties hold whp, for all $p \in S$.*

1. *If $|B_S(p, \lambda)| \geq 5\kappa$, then $b_{\lambda,\kappa}(p, S) = 1$.*

2. *If $|B_S(p, 2\gamma\lambda)| < \kappa$, then $b_{\lambda,\kappa}(p, S) = 0$.*

*Here, for any given $r > 0$, the symbol $B_S(p, r) := \{q \in S : \mathtt{dist}(p, q) \leq r\}$ denotes the* ball of radius $r$ around $p$. *The data structure has an amortized update time of $poly(d) \cdot n^{O(1/\gamma^2)}$, where $n = |S|$ is the size of the input point-set. The data structure works correctly with high probability and is robust against an adaptive adversary.*

**Corollary C.3.** For all $\lambda > 0$, there is a dynamic data structure $\mathcal{D}_\lambda(S)$ for handling a point-set $S$ going through a sequence of updates (point insertions/deletions). Upon receiving a *query*, the data structure returns a point $p \in S$ and a value $\kappa \geq 1$ such that

1. $|B_S(p, 2\gamma\lambda)| \geq \kappa$, and

2. for all points $q \in S$, we have $|B_S(q, \lambda)| < 10\kappa$.

Here, for any given $r > 0$, the symbol $B_S(p, r) := \{q \in S : \mathtt{dist}(p, q) \leq r\}$ denotes the *ball of radius $r$ around $p$*. The data structure has an amortized update time of $poly(d) \cdot n^{O(1/\gamma^2)}$, where $n = |S|$ is the size of the input point-set. The data structure works with high probability and is robust against an adaptive adversary.

*Proof.* We make $\Theta(\log n)$ possible *guesses* for the value of $\kappa$, in powers of 2. For each such guess $\kappa$, we maintain the data structure $\mathcal{D}_{\lambda,\kappa}(S)$ as guaranteed by Theorem C.2. Upon receiving a query, we identify the largest guess $\kappa$ such that $b_{\lambda,\kappa}(p, S) = 1$ for at least one point $p \in S$, and return the concerned point $p$ and the guess $\kappa$. $\square$

### C.2. Basic Building Block: Pseudo-Average Costs

Consider a set $F$ of facilities and a set $D$ of clients. Every facility $i \in F$ has opening cost $f_i > 0$. Given a regular cluster $C = (i, A)$, with $i \in F$ and $A \subseteq D$, we define its *pseudo-average cost* to be

$$\mathtt{avg\text{-}cost}^\star(C) = \frac{f_i + |A| \cdot \mathtt{radius}(C)}{|A|}.$$

In other words, while computing the pseudo-average cost of a regular cluster, we pretend that all its clients are situated at its boundary. This immediately leads to the following observation.

**Observation C.4.** For every regular cluster $C$ defined over $(F, D)$, we have $\mathtt{avg\text{-}cost}(C) \leq \mathtt{avg\text{-}cost}^\star(C)$. Let $\mathtt{Opt}(F, D)$ and $\mathtt{Opt}^\star(F, D)$ respectively be the minimum average cost and pseudo-average cost of any regular cluster defined over $(F, D)$. Then, $\mathtt{Opt}(F, D) \leq \mathtt{Opt}^\star(F, D)$.

**Claim C.5.** Let $C = (i, A)$ be a regular cluster defined over $(F, D)$ that has minimum average cost, i.e., $\texttt{avg-cost}(C) = \texttt{Opt}(F, D)$. Then, we must have $\texttt{avg-cost}^\star(C) \leq 2 \cdot \texttt{avg-cost}(C)$.

*Proof.* If $|A| = 1$, then it is easy to verify that $\texttt{avg-cost}^\star(C) = \texttt{avg-cost}(C)$. For the rest of the proof, we assume that $|A| > 1$. Let $\texttt{radius}(C) = r$. We consider two possible cases.

*Case 1.* $r \leq \texttt{avg-cost}(C)$. In this case, we infer that $\texttt{avg-cost}^\star(C) = f_i/|A| + r \leq \texttt{avg-cost}(C) + r \leq 2 \cdot \texttt{avg-cost}(C)$.

*Case 2.* $r > \texttt{avg-cost}(C)$. Let $j := \arg\max_{j' \in A}\{\text{dist}(i, j)\}$, so that $\text{dist}(i, j) = r = \texttt{radius}(C)$. Let $A' := A \setminus \{j\}$. Since $|A| > 1$, we have $A' \neq \emptyset$. Consider the regular cluster $C' := (i, A')$ obtained by removing the client $j$ from $C$. Now, we derive that $|A'| \cdot \texttt{avg-cost}(C') = |A| \cdot \texttt{avg-cost}(C) - \text{dist}(i, j) = |A| \cdot \texttt{avg-cost}(C) - r < (|A| - 1) \cdot \texttt{avg-cost}(C) = |A'| \cdot \texttt{avg-cost}(C)$. Diving both sides of the resulting inequality by $|A'|$, we get $\texttt{avg-cost}(C') < \texttt{avg-cost}(C)$. This leads to a contradiction, since we assumed that $C$ is a regular cluster defined over $(F, D)$ with minimum average cost. In other words, we can never end up in Case 2.

This concludes the proof of the claim. $\qquad\square$

**Corollary C.6.** We have $\texttt{Opt}(F, D) \leq \texttt{Opt}^\star(F, D) \leq 2 \cdot \texttt{Opt}(F, D)$.

*Proof.* Let $C = (i, A)$ be a regular cluster defined over $(F, D)$, with $\texttt{avg-cost}(C) = \texttt{Opt}(F, D)$. Then, we have $\texttt{Opt}^\star(F, D) \leq \texttt{avg-cost}^\star(C) \leq 2 \cdot \texttt{avg-cost}(C) = 2 \cdot \texttt{Opt}(F, D)$; the second inequality follows from Claim C.5. Finally, Observation C.4 gives us $\texttt{Opt}(F, D) \leq \texttt{Opt}^\star(F, D)$. $\qquad\square$

**Corollary C.7.** Consider any $\tau \geq 1$. Let $C = (i, A)$ be a regular cluster defined over $(F, D)$ with $\tau$-approximate minimum pseudo-average cost; i.e., $\texttt{avg-cost}^\star(C) \leq \tau \cdot \texttt{Opt}^\star(F, D)$. Then, $C$ also has $2\tau$-approximate minimum average cost; i.e., $\texttt{avg-cost}(C) \leq 2\tau \cdot \texttt{Opt}(F, D)$

*Proof.* From Observation C.4 and Corollary C.6, we infer that $\texttt{avg-cost}(C) \leq \texttt{avg-cost}^\star(C) \leq \tau \cdot \texttt{Opt}^\star(F, D) \leq 2\tau \cdot \texttt{Opt}(F, D)$. $\qquad\square$

### C.3. Cheapest Cluster Oracle: Proof of Lemma 2.10

We partition the set $F$ of facilities into $\Theta(\log m)$ groups $F_1, \ldots, F_{\Theta(\log m)}$, by discretizing their opening costs in powers of 2. Specifically, let $f_{\min} = \min_{i \in F}\{f_i\}$ denote the minimum opening cost among all facilities. Then every facility in group $F_k$, for $k \in [1, \Theta(\log m)]$, has opening cost in the range $[f_{\min} \cdot 2^{k-1}, f_{\min} \cdot 2^k)$. Next, we scale down the opening costs of facilities by at most a factor of 2, so that each facility $i \in F_k$ in group $k$ now has a scaled down opening cost of $\hat{f}_i = f_{\min} \cdot 2^{k-1}$. Since $\hat{f}_i \leq f_i \leq 2 \cdot \hat{f}_i$ for all facilities $i$, any regular cluster with (say) $\tau$-approximate minimum scaled down average cost will have an actual average cost that is $2\tau$-approximate (for any $\tau \geq 1$). From now on, we will focus on designing a cheapest cluster oracle on the given input $(F, D)$ w.r.t. the scaled down facility opening costs. This will incur at most an additional factor 2 loss in the approximation ratio. Furthermore, we will maintain a separate cheapest cluster oracle $(F_k, D)$ for each group $F_k$ of facilities. Note that a client $j \in D$ will participate in each of these $\Theta(\log m)$ oracles, one for each $k \in [1, \Theta(\log m)]$. In response to a query in the original input instance, we will make a query to each of these $\Theta(\log m)$ oracles, and return the best (i.e., with minimum average cost) answer among the ones returned by these oracles. This would incur an extra $\Theta(\log m)$ factor overhead in update and query times. To summarize, we infer that to prove Lemma 2.10, it suffices to focus on the special case where every facility in the input has the same opening cost. This is captured in the assumption below.

**Assumption C.8.** Every facility $i \in F$ has the same opening cost $f_i = f$ (say).

Armed with Corollary C.7, it suffices to focus on designing a dynamic data structure that, when given a solution-query (resp. value query), can return a regular cluster (resp. pseudo-average cost of a regular cluster) in $(F, D)$ with $\gamma$-approximate minimum pseudo-average cost. By Corollary C.7, such a cluster would also have $2\gamma$-approximate minimum average cost.

$r$**-Cheapest-Cluster Oracle.** Fix any parameter $r > 0$. We will now design an $r$-*cheapest-cluster oracle* on the input $(F, D)$, which works as follows. As per Assumption C.8, every facility $i \in F$ has the same opening cost (say) $f > 0$. The oracle can handle *updates* to the set of clients $D$, where each update inserts/deletes a client. Upon receiving a *value-query*,

the oracle returns a value $t = O(\gamma) \cdot \texttt{Opt}_r^\star(F, D)$ such that there exists a regular cluster $C$ defined over $(F, D)$ with $\texttt{avg-cost}^\star(C) \leq t$. Here, $\texttt{Opt}_r^\star(F, D)$ denotes the minimum pseudo-average cost of any cluster defined over $(F, D)$ that has radius in the range $[r/2, r)$. Similarly, upon receiving a *solution-query*, the oracle returns a regular cluster $C$ defined over $(F, D)$ such that $\texttt{avg-cost}^\star(C) = O(\gamma) \cdot \texttt{Opt}_r^\star(F, D)$. Note that the radius of the returned cluster $C$ here might lie outside the range $[r/2, r)$. We will show how to design such an $r$-cheapest cluster oracle with the $\texttt{poly}(d) \cdot (m + n)^{O(1/\gamma^2)}$ update time and value-query time, and $\texttt{poly}(d) \cdot (m+n)^{O(1/\gamma^2)} \cdot |A|$ solution-query time, where $|A|$ is the size of the cluster $C = (i, A)$ returned in response to a solution-query. This will imply Lemma 2.10, since we can make $\Theta(\log m)$ guesses for the value of the (approximate) radius $r$, and run an $r$-cheapest-cluster oracle for each such guess *in parallel* (i.e., using $\Theta(\log m)$ different and independent copies of the oracle), and upon receiving a query return the best answer we receive from the collection of these $\Theta(\log m)$ oracles.

**Set of Relevant Clients.** To implement an $r$-cheapest-cluster oracle, we maintain a subset $\hat{D} \subseteq D$ of *relevant clients* and a mapping $\phi : \hat{D} \to F$, which are defined as follows. We run a dynamic nearest-neighbor oracle on the set $F$ of facilities. Whenever a client $j$ is inserted in the set $D$, we make a query to the dynamic nearest-neighbor oracle to find a $\gamma$-approximate nearest neighbor of $j$ in $F$. Let $i \in F$ be the facility returned by the dynamic nearest-neighbor oracle in response to this query. We add $j$ to the set $\hat{D}$ and set $\phi(j) \leftarrow i$ iff $\text{dist}(i, j) < \gamma \cdot r$. Furthermore, whenever a client $j$ gets deleted from $D$, we set $\hat{D} \leftarrow \hat{D} \setminus \{j\}$. By Theorem C.1, it takes $\texttt{poly}(d) \cdot m^{O(1/\gamma^2)}$ time per update to maintain the set $\hat{D}$, and each update in $D$ leads to at most one update in $\hat{D}$. Finally, the set of relevant points $\hat{D} \subseteq D$ satisfy the following conditions.

1. For each client $j \in D$, if $\text{dist}(j, F) < r$, then $j \in \hat{D}$.

2. For each client $j \in \hat{D}$, we have $\text{dist}(j, F) \leq \text{dist}(j, \phi(j)) < \gamma \cdot r$.

**A Modified Input Instance.** We now consider a modified input instance $(\hat{F}, \hat{D})$, where $\hat{F} = \hat{D}$ (i.e., there is a facility at the location of each relevant client) and each facility $i \in \hat{F}$ has the same opening cost $f$, as before. In the claim below, we show that there is a useful correspondence between the regular clusters in the original input $(F, D)$ and the regular clusters in the modified input $(\hat{F}, \hat{D})$.

**Claim C.9.** Consider any regular cluster $C = (i, A)$ with $\texttt{radius}(C) \in [r/2, r)$ in the original input $(F, D)$. Then, we must have $A \subseteq \hat{D}$. Further, for every $j \in A$, the regular cluster $\hat{C}_j = (j, A)$ has $\texttt{radius}(\hat{C}_j) < 2 \cdot r$ in the modified input $(\hat{F}, \hat{D})$.

*Proof.* For all $j' \in A$, we have $\text{dist}(j', F) \leq \text{dist}(i, j') \leq \texttt{radius}(C) < r$. Hence, we get $A \subseteq \hat{D}$. Next consider any point $j \in A$ and the regular cluster $\hat{C}_j = (j, A)$ in the modified input $(\hat{F}, \hat{D})$. For every point $j' \in A$, we have $\text{dist}(j, j') \leq \text{dist}(j, i) + \text{dist}(i, j') \leq 2 \cdot \texttt{radius}(C)$. Thus, we infer that $\texttt{radius}(\hat{C}_j) \leq 2 \cdot \texttt{radius}(C) < 2 \cdot r$.  □

Armed with Claim C.9, our $r$-cheapest-cluster oracle works as follows. We run the data structure $\mathcal{D}_{2r}(\hat{D})$, as in Corollary C.3, on the dynamic point-set $\hat{D}$. When we receive a value-query, we make a query to $\mathcal{D}_{2r}(\hat{D})$, which returns a client $j \in \hat{D}$ and a value $\kappa \geq 1$ such that

1. $|B_{\hat{D}}(j, 4\gamma r)| \geq \kappa$, and

2. for all clients $j' \in \hat{D}$, we have $|B_{\hat{D}}(j', 2r)| < 10\kappa$.

In response to the value-query, we return the value $t = f/\kappa + 9\gamma r$. The correctness of this procedure follows from the claim below.

**Claim C.10.** We have $t = O(\gamma) \cdot \texttt{Opt}_r^\star(F, D)$. Furthermore, there exists a regular cluster $C' = (i', A')$ defined over $(F, D)$ with $\texttt{avg-cost}^\star(C') \leq t$.

*Proof.* Let $C = (i, A)$ be a regular cluster in $(F, D)$ with $\texttt{radius}(C) \in [r/2, r)$ such that $\texttt{avg-cost}^\star(C) = \texttt{Opt}_r^\star(F, D)$. By Claim C.9, we have $A \subseteq \hat{D}$. Fix any client $j \in A$, and consider the regular cluster $\hat{C}_j = (j, A)$ in the modified input $(\hat{F}, \hat{D})$. By Claim C.9, we have $\texttt{radius}(\hat{C}_j) < 2r$. Next, recall that $|B_{\hat{D}}(j', 2r)| < 10\kappa$ for all $j' \in \hat{D}$. Accordingly, we infer that $|A| < 10\kappa$, and hence $\texttt{Opt}_r^\star(F, D) = \texttt{avg-cost}^\star(C) = f/|A| + \texttt{radius}(C) > f/(10\kappa) + \texttt{radius}(C) \geq f/(10\kappa) + r/2$. Since $t = f/\kappa + 9\gamma r$, it follows that $t = O(\gamma) \cdot \texttt{Opt}_r^\star(F, D)$.

Next, consider the cluster $C' = (i', A')$, where $A' := B_{\hat{D}}(j, 4\gamma r)$ and $i' := \phi(j)$. Observe that $|A'| \geq \kappa$. For every client $j' \in A'$, we get $\texttt{dist}(i', j') \leq \texttt{dist}(i', j) + \texttt{dist}(j, j') = \texttt{dist}(\phi(j), j) + \texttt{dist}(j, j') \leq \gamma r + \texttt{dist}(j, j') \leq \gamma r + 8\gamma r = 9\gamma r$, where the last inequality holds since any two points within the ball $B_{\hat{D}}(j, 4\gamma r) = A'$ are within at most a distance of $4\gamma r + 4\gamma r = 8\gamma r$ away from each other. It follows that $\texttt{radius}(C') \leq 9\gamma r$, and hence $\texttt{avg-cost}^\star(C') = f/|A'| + \texttt{radius}(C') \leq f/\kappa + \texttt{radius}(C') \leq f/\kappa + 9\gamma r = t$. □

In response to a solution-query, we first perform the same steps as in a value-query. This gives us a client $j \in \hat{D}$ and a value $\kappa \geq 1$ such that

1. $|B_{\hat{D}}(j, 4\gamma r)| \geq \kappa$, and

2. for all clients $j' \in \hat{D}$, we have $|B_{\hat{D}}(j', 2r)| < 10\kappa$.

We now keep making queries to the dynamic nearest-neighbor oracle on the point-set $\hat{D}$, to repeatedly find a ($\gamma$-approximate) nearest neighbor of $j$ in $\hat{D} \setminus \{j\}$ (and after receiving a client $j'$ in response to the query we remove $j'$ from $\hat{D}$; we add all these removed clients back to $\hat{D}$ at the end of this process). This allows us to recover (an approximation of) the cluster $C' = (i', A')$ as in the proof of Claim C.10, in time proportional to $|A'|$, modulo a multiplicative factor of $\text{poly}(d) \cdot n^{O(1/\gamma^2)}$. We then return the cluster $C'$ in response to the solution-query.

The update time, value-query time and solution-query time of this procedure follow from the guarantees of Theorem C.1 and Corollary C.3.

### C.4. Bi-Chromatic Closest Pair Oracle: Proof of Lemma 2.9

Fix any parameter $r > 0$. We first define an $r$-**bi-chromatic closest pair oracle**, denoted by $r - \texttt{Closest-Pair}(F, D, \gamma)$, as follows. Here, $F$ is a set of facilities, $D$ is a set of clients, and $\gamma \geq 1$. We refer to $\gamma$ as the approximation ratio of the oracle. The oracle supports the following operations.

- UPDATE: Insert/delete a point in the set $F$ or $D$.

- QUERY: In response to a query, the oracle either returns an ordered pair $(p, q) \in F \times D$ with $\text{dist}(p, q) \leq \gamma r$, or it returns $\perp$. If the oracle returns $\perp$, then it is guaranteed that $\min_{(p', q') \in F \times D} \{\text{dist}(p', q')\} \geq r$.

**Lemma C.11.** For every $r > 0$ and every sufficiently large absolute constant $\gamma > 1$, there exists an implementation of an $r$-bi-chromatic closest pair oracle with $\gamma$ approximation ratio, which handles queries and updates in $\text{poly}(d) \cdot (m+n)^{O(1/\gamma^2)}$ time. The oracle works correctly with high probability and is robust against an adaptive adversary.

It is easy to verify that Lemma C.11 implies Lemma 2.9. This is because we can guess the value of $r$ in powers of 2; there are $\Theta(\log m)$ many such guesses. For each such guess $r$, we maintain the oracle $r - \texttt{Closest-Pair}(F, D, \gamma)$. Taken together, these $\Theta(\log m)$ oracles allow us to construct a oracle for $\texttt{Closest-Pair}(F, D, \Theta(\gamma))$. So, we devote the rest of this section to the proof of Lemma C.11.

In essence, we maintain a maximal matching between the facilities and the clients in an appropriately chosen *threshold graph*, where a facility and a client are connected by an edge if they are sufficiently close to each other. Let this maximal matching be $M$. We also explicitly maintain the matched facility-client pairs in $M$. Let $F(M) \subseteq F$ and $D(M) \subseteq D$ respectively denote the sets of facilities and clients matched under $M$. We run a dynamic nearest neighbor oracle on each of the point-sets $F \setminus F(M)$ and $D \setminus D(M)$.

Now, consider the insertion of a client $j$. To handle this insertion, we make a query to the nearest-neighbor oracle for $F \setminus F(M)$ w.r.t. $j$. Let $i \in F \setminus F(M)$ be the facility returned in response to this query. If $\text{dist}(i, j) \leq \gamma r$, then we add the edge $(i, j)$ to the matching $M$, add the facility $i$ to the set $F(M)$ (which leads to the deletion of $i$ from the nearest-neighbor oracle for $F \setminus F(M)$), and add the client $j$ to the set $D(M)$. Otherwise, if $\text{dist}(i, j) > \gamma r$, then the matching $M$ remains unchanged, and we insert the client $j$ in the nearest-neighbor oracle for $D \setminus D(M)$.

Next, consider the deletion of a client $j$. If $j \notin D(M)$, then we simply delete the client $j$ from the nearest-neighbor oracle for $D \setminus D(M)$. Otherwise, we have $j \in D(M)$, and let $i \in F(M)$ be the unique facility such that $(i, j) \in M$. In this case, we first delete the edge $(i, j)$ from the matching $M$. After that, we make a query to the nearest-neighbor oracle for

$D \setminus D(M)$ w.r.t. $i$. Let $j' \in D \setminus D(M)$ be the client returned in response to this query. If $\text{dist}(i, j') \leq \gamma r$, then we add the edge $(i, j')$ to the matching $M$ (which leads to the deletion of $j'$ from the nearest-neighbor oracle for $D \setminus D(M)$). Otherwise, we have $\text{dist}(i, j') > \gamma r$, and in this case we add the facility $i$ to the nearest-neighbor oracle for $F \setminus F(M)$.

Insertion/deletion of a facility in $F$ is handled in a completely analogous manner.

From the above description and the approximation guarantee of the dynamic nearest-neighbor oracle (see Theorem C.1), it is easy to verify that the matching $M$ satisfies the following two properties. (i) $\text{dist}(i, j) \leq \gamma r$ for all $(i, j) \in M$. (ii) $\text{dist}(i, j) > r$ for all $i \in F \setminus F(M)$ and $j \in D \setminus D(M)$.

Upon receiving a query for the oracle $r - \text{Closest-Pair}(F, D, \gamma)$, we perform the following operations. If the matching $M$ is non-empty, then we return any arbitrary pair $(i, j) \in M$ in response to the query. Otherwise, if $M = \emptyset$, then we return $\perp$. It is easy to verify that this satisfies the guarantees as stated in Lemma C.11.

Each update in $(F, D)$ leads to at most constantly many updates and queries in the dynamic nearest-neighbor oracles for the sets $F \setminus F(M)$ and $D \setminus D(M)$. Accordingly, the update and query time guarantees of Lemma C.11 follow from Theorem C.1.

