# OpenReview forum: "Dynamic High-Dimensional Facility Location with Low Recourse"
_ICML.cc/2026/Conference — ICML 2026 regular_

### Official Review · Reviewer_5QkF · 2026-02-17

**Soundness:** 3
**Presentation:** 3
**Significance:** 3
**Originality:** 3
**Overall Recommendation:** 5
**Confidence:** 4

**Summary:**

The paper studies the dynamic facility location problem in $d$-dimensional Euclidean spaces, and presents an algorithm that maintains a $\gamma$-approximate solution with $O(\log m)$ amortized recourses and $\mathrm{poly}(d) \cdot (m+n)^{O(1/\gamma)}$ amortized update time for some sufficiently large constant $\gamma$.  The algorithm guarantees the approximation ratio by maintaining an $(\alpha, \beta, \gamma)$-partition derived from the greedy UFL algorithm. The recourse bound and update time are analyzed through the movement-cost: The paper shows that the number of recourses is bounded by the total movement-cost and that the update time is essentially bounded by the movement-cost multiplied by the time of queries to the bi-chromatic closest pair oracle and the cheapest cluster oracle. The total movement-cost is then bounded using a token argument exploiting the fact that the level of a client increases only when the regular cluster to which the client is assigned loses a sufficient number of clients assigned to the cluster center. The two oracles are built with a dynamic nearest-neighbor oracle for $d$-dimensional Euclidean spaces and a tailored doubling scheme.

**Compliance With Llm Reviewing Policy:**

Affirmed.

**Final Justification:**

My concerns had been adequately addressed. I will maintain my initial positive score.

**Key Questions For Authors:**

1. Can the result extend to other settings such as the capacitated or prize-collecting/robust settings? My naive hunch says that the paper's approach may smoothly extend to the prize-collecting/robust setting as it's well-known JMS-like algorithms work well in the setting. Not so sure with the capacitated case, though.

See also Strengths and Weaknesses.

**Limitations:**

See Strengths and Weaknesses.

**Strengths And Weaknesses:**

The problem considered by this paper is well motivated and relevant in machine learning and clustering. The algorithm design and its analysis are nontrivial and solid; I found some errors (to be stated), but they look easily fixable. The paper is also generally well written and easy to follow; some steps are somewhat too abstract for me, requiring me some time to convince myself, but I think it’s acceptable. Given these merits, I believe the paper would attract a fair amount of attention from audiences, especially from the fields of clustering and dynamic algorithms.

One major concern is in handling the deletion of a client $j$. Currently, the algorithm calls $\text{Remove-Client}(C, j)$ where $C$ is the cluster to which $j$ is assigned, followed by a call to $\text{Fix-Invariant}()$. To my understanding, it is intended to have Invariants 2.1, 2.2, and 2.3 all satisfied at the start of every call to $\text{Fix-Invariant}$, but observe that a call to $\text{Remove-Client}(C, j)$ can make $C$ violate Invariant 2.3. However, I think it is easily fixable by calling $\text{Move-Up}(C)$ with the same $C$ in this case.

I have more comments on concerns, typos, or presentations in the paper. Including the above concern, these should be properly addressed before publication of this paper.
1. l63, col1: Is it without loss of generality to assume that the distances and facility opening costs are polynomially bounded by $m$?
2. l102, col1, Remarks: Isn’t the update time of the proposed algorithm is amortized $\textrm{poly}(d) \cdot (m+n)^{O(1/\gamma^2)}$? Isn’t it better than having $1/\gamma$ on the exponent?
3. l55, col2: $O(m+n)^{1/\gamma}$ &rarr; $(m+n)^{O(1/\gamma)}$?
4. l165, col2: It might be better to use other symbols than $F$ and $D$, as they are already used for the facility and client sets of the ground FL instance. Or, clearly flag this duplicate usage.
5. l181, col2, Lem2.9: Remove $r$, as it is defined in Appendix C.4.
6. l234, col1: It would be more instructive to explicitly mention that there can be more than one regular clusters with the same center. I was a bit confused when I first read this part because I had a false impression that every facility can be assigned to at most one regular cluster.
7. l239, col1: (resp. $C \in \mathcal{C}^\star_i(k)$) &rarr; (resp. $C \in \mathcal{C}_i(k)$)
8. l243, col1: $\gamma$ is missing in the argument set of each oracle.
9. l252, col1: $\text{Cheap-Cluster}(F^\star, D_{>k}, \gamma)$ &rarr; $\text{Cheap-Cluster}(F, D_{>k}, \gamma)$
10. l379, col1: from a being part &rarr; from being part
11. l499, col1: Jain et al. (2003) is duplicate.
12. l576: which satisfy the &rarr; which satisfies the
13. l606, LemA.4: Do we need $\alpha$ in the upper bound on $v_j$? More precisely, isn’t it possible to have $v_j \leq 2 \gamma (\beta + 1) v_{j_1} + 2 \gamma \cdot dist(i, j)$?
14. l611: $i_i$ &rarr; $i_1$
15. l777: $2 \texttt{avg-cost}^\star(C)$ &rarr; $2 \texttt{avg-cost} (C)$
16. l832: What do you mean by "guess in parallel"?
17. l845: $\textrm{dist}(j, \phi(j)) \leq \textrm{dist}(j, F)$ &rarr; Swap the sides?
18. l858: $\leq \text{dist}(j, i)) +$ &rarr; Remove the redundant ")"
19. l923: We we run &rarr; We run

---

> ### Author Rebuttal · Authors · 2026-03-30
>
> ### Concern 4.1: Handling Deletion of a Client (Invariant 2.3)
> > A call to delete a client can make the cluster violate Invariant 2.3 without triggering a repair step. Can this be fixed by calling the repair step with the same cluster in this case?
>
> We thank the reviewer for the suggested fix. Please see our detailed response to Reviewer f7MX (Concern 1.1) regarding the maintenance of Invariant 2.3 after deletions.
>
> ### Concern 4.2: Extension to Capacitated or Prize-Collecting/Robust Settings
> > Can the result extend to other settings such as the capacitated or prize-collecting/robust settings?
>
> We thank the reviewer for raising this question. We agree that most likely a similar approach might work for the prize collecting variant; whereas dealing with capacitated settings would probably require fundamentally new insights. However, we treat these as exciting research questions to pursue as follow up work, and we are unable to resolve these within the limited time period of the rebuttal phase.
>
>
>
> ### Concern 4.3: Line-by-line comments on notation and write-up
>
> We will address all of the typos and notation inaccuracies in the updated version of our paper. We answer the questions raised below:
>
> ### Comment 1
> > l63, col1: Is it without loss of generality to assume that the distances and facility opening costs are polynomially bounded by $m$?
>
> Assuming that the distances and facility opening costs are polynomially bounded is a very standard assumption in clustering problems. The reason is that many algorithms and techniques naturally lead to an $O(\log \Delta)$ term in the running time or recourse, where $\Delta$ is the aspect ratio (ratio of smallest non-zero to largest opening cost/distance). This assumption simply allows us to replace these $\log \Delta$ terms with $\log m$, simplifying our expressions. We note that some lines of work on clustering focus on obtaining better dependence on $\Delta$ (such as $\log \log \Delta$), implicitly considering settings where $\Delta$ can be very large, but this is not the focus of our paper.
>
> ### Comment 2
> > l102, col1, Remarks: Isn't the update time of the proposed algorithm  amortized $\text{poly}(d) \cdot (m+n)^{O(1/\gamma^2)}$? Isn't it better than having $1/\gamma$ on the exponent?
>
> Kindly note that while our update time bound (as stated in Lemma 2.20) is indeed $\text{poly}(d) (m+n)^{1/\gamma^2}$, our approximation ratio (as stated in Theorem 2.8) is $O(\gamma^2)$. Thus, replacing $\gamma^2$ by $\gamma$, we get the guarantee of Theorem 1.1. In the revised version of the paper, we will add a remark to explicitly clarify this point.
>
> ### Comment 3
> > l55, col2: $O(m+n)^{1/\gamma} \rightarrow (m+n)^{O(1/\gamma)}$?
>
> Many thanks for pointing this out. We will correct it in the revised version.
>
> ### Comment 13
> > l606, LemA.4: Do we need $\alpha$ in the upper bound on $v_j$? More precisely, isn't it possible to have $v_j \le 2\gamma(\beta+1)v_{j_1} + 2\gamma \cdot \text{dist}(i, j)$?
>
> We sincerely thank the reviewer for pointing this out. Indeed, this extra $\alpha$ factor is not needed. Kindly also check our response to Concern 1.2 by Reviewer f7MX.
>
> ### Comment 16
> > l832: What do you mean by "guess in parallel"?
>
> We use the phrase ‘in parallel’ to emphasise that we are using $\Theta(\log m)$ different and independent copies of this oracle, and thus we can handle the queries to these oracles independently. In the revised version of the paper, we will add a remark to clarify this point.
>
> ### Comment 17:
> > l845: $\text{dist}(j, \phi(j)) \le \text{dist}(j, F) \rightarrow$ Swap the sides?
>
> Indeed, we need to swap the sides of this inequality. Many thanks for pointing out this typo.

---

> > ### Author Rebuttal · Reviewer_5QkF · 2026-04-01
> >
> > Thank you for the response. My concerns have been adequately addressed.

---

### Official Review · Reviewer_FRd8 · 2026-03-07

**Soundness:** 3
**Presentation:** 2
**Significance:** 3
**Originality:** 3
**Overall Recommendation:** 5
**Confidence:** 3

**Summary:**

This study addresses the Dynamic Facility Location problem with non-uniform costs in high-dimensional Euclidean spaces. The authors propose a randomized algorithm that simultaneously achieves a γ-approximation, O(logm) amortized recourse, and poly(d)⋅(m+n)O(1/γ) amortized update time for a sufficiently large constant γ≥1. The core idea is to dynamize a static greedy algorithm to maintain a level-based partition of clients. By dropping a restrictive invariant present in prior works and introducing novel data structures tailored for high-dimensional spaces—such as the Bi-chromatic Closest Pair Oracle and the Cheapest Cluster Oracle.

**Compliance With Llm Reviewing Policy:**

Affirmed.

**Final Justification:**

Since authors appropriately responded to my concerns, I keep my original positive review.

**Key Questions For Authors:**

- Could the proposed framework be extended to other metric families  where approximate NN search is possible, or does the analysis fundamentally rely on Euclidean geometric properties?
- To establish the optimal triple trade-off among approximation, recourse, and update time, is it possible to prove a theoretical lower bound demonstrating that an update time of o((m+n)1/γ) inherently requires an Ω(logm)recourse?

**Limitations:**

- To bypass the constraints of prior methods, the algorithm allows for opening multiple clusters at the exact same facility location. This limits its direct applicability to real-world scenarios where a physical facility strictly cannot be opened more than once.
- Although the poly(d) dependence can theoretically be reduced to O(log(m+n)) via Johnson-Lindenstrauss dimensionality reduction, the strict reliance on LSH-based techniques means the algorithm is currently bounded by the technological limits of LSH, rather than achieving an absolute theoretical optimal.

**Strengths And Weaknesses:**

### Strength

- It provides the first efficient dynamic algorithm for the facility location problem with non-uniform costs in high-dimensional Euclidean spaces.
- It proves a rigorous and same recourse but better update time compared to existing dynamic solutions.
- The paper introduces a token-based accounting method (bank-account method) to tightly analyze the movement-cost associated with cluster level changes.
- The algorithm guarantees correct execution with high probability even against adaptive adversaries.

### Weakness

- The approximation ratio γ is defined as a sufficiently large constant. Attempting to decrease γ for better accuracy exponentially worsens the update time, meaning the practical approximation precision might be limited.
- While the paper argues that improving the results would require breaking current LSH barriers, it does not provide true information-theoretic or computational hard lower bounds to prove that the current trade-off is optimal.

---

> ### Author Rebuttal · Authors · 2026-03-30
>
> ### Concern 3.1: Approximation Ratio Constant and Update Time
> > Attempting to decrease $\gamma$ for better accuracy exponentially worsens the update time, meaning the practical approximation precision might be limited.
>
> Please see our response to Concern 1.5 of reviewer f7MX
>
> ### Concern 3.2: Lower Bounds for the Optimal Trade-off
> > To establish the optimal triple trade-off among approximation, recourse, and update time, is it possible to prove a theoretical lower bound demonstrating that an update time of $o((m+n)^{1/\gamma})$ inherently requires an $\Omega(\log m)$ recourse?
>
> We find this to be a very interesting future research direction. Even if we completely ignore the update time of an algorithm in high dimensional Euclidean space, it remains a challenging open question to get constant approximation with constant recourse when the facility opening costs can be non-uniform (in general metric space). Indeed, the previous relevant paper in this context, by  [Bhattacharya, Lattanzi, Parotsidis; NeurIPS 2022], also had logarithmic recourse. In the revised version of the paper, we will explicitly point this out as an interesting open question.
>
> ### Concern 3.3: Extension to Other Metric Families
> > Could the proposed framework be extended to other metric families where approximate NN search is possible, or does the analysis fundamentally rely on Euclidean geometric properties?
>
> Please see footnote 1 in the introduction of our paper: ‘Our results are not restricted to the Euclidean setting and apply to any metric spaces admitting efficient dynamic Nearest Neighbor (NN) Oracle access. Although in the presentation we focus on Euclidean metric for sake of simplicity’
>
> ### Concern 3.4: Applicability Due to Multiple Clusters at One Facility
> > Allowing for opening multiple clusters at the exact same facility location limits direct applicability to real-world scenarios where a physical facility strictly cannot be opened more than once.
>
> Thank you for this comment. We would like to emphasize that the notion of opening multiple clusters at the same facility is an artifact of the analysis (and so, even though we may pay a facility opening cost multiple times, the theoretical guarantees still hold). A practical implementation can easily avoid doing this and actually open a new facility $f$ only when the theoretical algorithm opens $f$ *for the first time* (and similarly, close $f$ when the last facility at $f$ is closed).
>
> ### Concern 3.5: LSH Technological Limits vs. Theoretical Optimal
> > The strict reliance on LSH-based techniques means the algorithm is currently bounded by the technological limits of LSH, rather than achieving an absolute theoretical optimal via Johnson-Lindenstrauss.
>
> We acknowledge this limitation. More broadly, our formulation of dynamic facility location essentially generalizes the approximate nearest neighbor search problem, and so the update time of our algorithm is lower bounded by the running time of running nearest neighbor search queries.
> To see this, given a point set $P$, we can construct a facility location instance, where the set of facilities is $P$ and the cost of opening each facility is $1/\text{poly}(|P|)$. Then, in order to issue a query asking for the nearest neighbor of a point $q$ in $P$, we insert a client at $q$ (that gets deleted right after a query). An approximate solution to this facility location instance approximately answers the approximate nearest neighbor query.

---

> > ### Author Rebuttal · Reviewer_FRd8 · 2026-04-03
> >
> > Thanks for your sincere rebuttal. Based on it, I keep my original positive evaluation.

---

### Official Review · Reviewer_MJny · 2026-03-13

**Soundness:** 3
**Presentation:** 3
**Significance:** 3
**Originality:** 3
**Overall Recommendation:** 4
**Confidence:** 3

**Summary:**

The paper studies the **dynamic non-uniform facility location problem** in high-dimensional Euclidean spaces. In this problem, there is a fixed set of possible facilities, each with its own opening cost, and a set of clients that changes over time through insertions and deletions. After each update, the algorithm must maintain a solution that approximately minimizes the total **opening cost plus connection cost**, where the opening cost is the cost of the facilities that are opened and the connection cost is the total distance from each client to the facility serving it.

The paper focuses on achieving good guarantees with limited **resources**, where resource means the number of changes in the assignments of clients to facilities after each update. The main result is a randomized dynamic algorithm for high-dimensional Euclidean space that, for every sufficiently large constant $\gamma \ge 1$, maintains a $\gamma$-competitive solution, uses only $O(\log m)$ amortized resources per update, and has amortized update time $\mathrm{poly}(d)\cdot (m+n)^{O(1/\gamma)}$, where $m$ is the number of facilities, $n$ is the number of current clients, and $d$ is the dimension. The guarantee holds against an adaptive adversary and succeeds with high probability.

**Compliance With Llm Reviewing Policy:**

Affirmed.

**Key Questions For Authors:**

Could you clarify how large $\gamma$ needs to be for the theorem to hold?

**Strengths And Weaknesses:**

**Strengths:**
1. The paper studies a fundamental problem in unsupervised learning and dynamic algorithms, namely dynamic non-uniform facility location, which is clearly of interest to the ICML community.

2. The result is quite strong. As noted in the remark after Theorem 1.1, improving the trade-off achieved in the paper would likely require a major breakthrough in locality-sensitive hashing (LSH). This suggests that the paper is pushing close to the current frontier of what is achievable with existing techniques.

3. From an originality and technical perspective, the paper appears to contain substantial new ideas. Compared to the prior work of Bhattacharya et al. (2022) for general metric spaces, the authors exploit the structure of Euclidean space to obtain a different trade-off. In particular, they remove a key invariant used in the earlier dynamic algorithm, which changes the structure of the maintained solution, and they also introduce two new high-dimensional data-structural ingredients: a bi-chromatic closest-pair oracle and a cheapest-cluster oracle. Overall, I found the work technically deep and quite novel.

**Weaknesses:**
1. One limitation is that, although the theorem is asymptotically strong, it is less clear how close the result is to practical applicability. In particular, the guarantee is stated only for every sufficiently large constant $\gamma$, and the paper does not provide a concrete threshold for how large $\gamma$ must be. Combined with the lack of any empirical evaluation, this makes it difficult to assess whether the algorithm would be practical in realistic regimes.

2. The paper would benefit from a careful proofreading pass. For example:
   - “various computational model” should be “various computational models”
   - “capture real-world scenario” should be “capture real-world scenarios”
   - “by-pass” should be “bypass”
   - “require new ideas” should be “requires new ideas”

---

> ### Author Rebuttal · Authors · 2026-03-30
>
> ### Concern 2.1: Practical Applicability and the Large Constant
> > Could you clarify how large $\gamma$ needs to be for the theorem to hold, and discuss the practical applicability given the lack of empirical evaluation?
>
> Please see our response to Concern 1.5 of reviewer f7MX
>
> ### Concern 2.2: Minor Typos and Proofreading
> > The paper would benefit from a careful proofreading pass to fix minor grammatical errors and typos (e.g., "various computational model", "by-pass").
>
> Thank you, we will carefully spell-check the final manuscript before submitting it.

---

> > ### Author Rebuttal · Reviewer_MJny · 2026-04-03
> >
> > Thank you for the response. I maintain my score.

---

### Official Review · Reviewer_f7MX · 2026-03-13

**Soundness:** 2
**Presentation:** 2
**Significance:** 3
**Originality:** 3
**Overall Recommendation:** 4
**Confidence:** 3

**Summary:**

This paper studies dynamic facility location with non-uniform opening costs in high-dimensional Euclidean spaces. The main result is a randomized dynamic algorithm with a $\gamma$-approximation, $O(\log m)$ amortized recourse, and amortized update time $\mathrm{poly}(d)\cdot (m+n)^{O(1/\gamma)}$ against an adaptive adversary. The main technical idea is to replace the exact distance-based invariant used in prior work with a hierarchy of regular and satellite clusters, which makes it possible to use dynamic geometric subroutines such as bi-chromatic closest-pair and cheapest-cluster data structures.

**Compliance With Llm Reviewing Policy:**

Affirmed.

**Final Justification:**

My main concerns have been addressed. Accordingly, I am increasing my overall score.

**Key Questions For Authors:**

1. After deletions, how is Invariant 2.3 guaranteed or restored if no level becomes dirty under Invariants 2.4/2.5?
2. In Lemma A.4, could the authors justify the $\alpha\beta$ factor in the last inequality, or correct the constant used in Theorem 2.8?
3. Could the authors summarize the non-uniform opening-cost reduction from Appendix C in the main text and state its exact complexity overhead more explicitly?

**Limitations:**

The authors should more concretely discuss the method’s assumptions and applicability boundaries, its practical time and recourse overhead, and the potential fairness or coverage disparities that facility-location style deployment decisions may introduce in real-world settings.

**Strengths And Weaknesses:**

## Strengths
1. The paper takes a meaningful step beyond prior dynamic FL frameworks. The paper identifies a real limitation in earlier dynamic facility location methods: the standard level-distance invariant is too rigid for the Euclidean setting here. Relaxing this structure and allowing multiple clusters at one facility is a meaningful idea.
2. The recourse analysis is technically solid. The token-based amortized analysis is one of the better parts of the paper. The breakdown into down-cost, up-cost, and switch-cost gives a fairly clean way to track recourse in a complicated dynamic process.
3. The geometric subroutines are interesting. The closest-pair and cheapest-cluster components are technically nontrivial, especially under adaptivity. Some of these ideas may be useful beyond this specific problem.

## Weaknesses
1. The maintenance procedure appears to have a correctness gap. The main issue is with Algorithm 2 $FIX-INVARIANTS$. The outer $while$ loop runs only when there is a dirty level, i.e., when Invariant 2.4 or 2.5 is violated. After a deletion, however, the paper only argues that Invariants 2.1 and 2.2 still hold; it does not show that Invariant 2.3 is automatically preserved. This matters because the repair step for Invariant 2.3 is nested inside the dirty-level loop. So if a deletion causes a regular cluster to violate Invariant 2.3 but does not create a dirty level, then the repair step is never executed. The current draft does not seem to rule this out. This looks like a real gap in the maintenance argument and may affect the approximation guarantee built on these invariants.
2. The proof of the approximation factor is not fully convincing as written. Appendix A seems to contain a real issue in Lemma A.4, where an extra $\alpha$ factor appears to be introduced without support from the previous inequalities. Since this term is then used in the dual-fitting argument, the constant in Theorem 2.8 does not seem to follow cleanly from the current proof. This may be fixable, and the high-level result may still be true. But for a paper whose main contribution is a quantitative approximation guarantee, this is not a minor detail.
3. The treatment of non-uniform opening costs is too hidden in the appendix. A key part of the algorithm is pushed into Appendix C. In the main text, the update procedure is presented as if one only maintains a $Cheap-Cluster(F, D_{>k}, \gamma)$ oracle. But in the non-uniform opening-cost setting, the actual implementation is more involved: facilities are grouped by cost scale, multiple oracle instances are maintained, and their outputs are combined. This is not just an implementation detail. It is a central reduction for the non-uniform case, and it should be stated more clearly in the main text.
4. The claim about removing the $\mathrm{poly}(d)$ factor via on-the-fly JL is too strong. The introduction says that the $\mathrm{poly}(d)$ dependence can be removed by applying Johnson--Lindenstrauss reduction on the fly, even against an adaptive adversary. But the paper does not provide a proof or a precise citation for this statement in the dynamic adaptive setting. This does not break the main theorem, since the theorem itself still includes the $\mathrm{poly}(d)$ factor. Still, the remark is stronger than what is actually supported in the paper and should be removed or stated more carefully.

---

> ### Author Rebuttal · Authors · 2026-03-30
>
> ### Concern 1.1: Maintenance Correctness Gap (Algorithm 2 & Invariant 2.3)
> > After deletions, how is Invariant 2.3 guaranteed or restored if no level becomes dirty under Invariants 2.4/2.5?
>
> We sincerely thank the reviewer for pointing this out. There is, however, an easy fix, which works, as follows. In the subroutine for handling a client deletion (Algorithm 1), we add one extra step, just after the end of the nested if block (which takes effect when $A = \emptyset$). This extra step says that if $A \neq \emptyset$, then we repeatedly call the subroutine Move-Up(C) as long as C violates Invariant 2.3. Now, it is self-evident that all the invariants are restored by our algorithm. Fortunately, the analysis remains completely unchanged, and the same guarantee on the update time and recourse continues to hold. This is because the up-costs are accounted for in our analysis using Claim 2.22 and Claim 2.23; and both these claims continue to hold with the modifications. We will incorporate this change in the revised version of the paper.
>
> ### Concern 1.2: Approximation Factor Proof (Lemma A.4)
> > In Lemma A.4, could you justify the αβ factor in the last inequality, or correct the constant used in Theorem 2.8?
>
> We sincerely thank the reviewer for pointing this out. The inequality as stated is technically correct, since $\alpha$ is at least one. However, you are right that this extra $\alpha$ factor is not needed. As a consequence, the approximation ratio in Theorem 2.8 actually improves, since we can replace the expression $(\alpha \beta + 1)$ by $(\beta+1)$. We will incorporate this change in the revised version of the paper.
>
> ### Concern 1.3: Non-uniform Opening Costs Presentation
> > Could you summarize the non-uniform opening-cost reduction from Appendix C in the main text and state its exact complexity overhead more explicitly?
>
> Thank you for this comment. This reduction increases the update and query time of the cheapest cluster oracle by a $\Theta(\log m)$ factor. We will add a summarized version of the reduction to the main part of the paper.
>
> ### Concern 1.4: Johnson-Lindenstrauss Claim
> > The claim about removing the  factor via on-the-fly JL is stronger than what is actually supported in the paper and should be removed or stated more carefully.
>
> We are very grateful for this comment. Indeed, we agree that when we use JL to reduce dimensionality to logarithmic, our algorithm no longer works against an adaptive adversary. We apologize for the oversight in phrasing the remark, which we will clarify.
>
> ### Concern 1.5: Method Assumptions and Applicability Boundaries
> > Please concretely discuss the method’s assumptions, applicability boundaries, practical time/recourse overhead, and potential fairness or coverage disparities in real-world settings.
>
> Our focus in this paper is a theoretical analysis of algorithms for dynamic facility location. The problem is closely related to the nearest neighbor search problem, and so we make heavy use of approximate nearest neighbor search data structures. The need for sufficiently large $\gamma$ in the analysis comes directly from applying an existing approximate nearest-neighbor search (ANNS) data structure.
>
> The unfortunate reality of the ANNS field is that there is a visible disconnect between the methods that come with strong theoretical guarantees and methods that provide highly efficient updates and queries in practice. For the former, locality sensitive hashing (LSH) has been the primary source of theoretical guarantees. In contrast, for the latter, graph-based indices have demonstrated empirical performance far surpassing the performance predicted by the theoretical analysis of LSH (see e.g. ParlayANN: Scalable and Deterministic Parallel GraphBased Approximate Nearest Neighbor Search Algorithms, PPoPP’24).
>
> We believe that the ideas from our paper combined with a state-of-the-art graph index for ANNS queries may give a practical method for solving the dynamic facility location problem. While the approximation ratio of our theoretical algorithm is admittedly a large constant, this should be attributed primarily to the use of an ANNS data structure with theoretical guarantees. We expect that using a more practical ANNS data structure, one that does not come with strong theoretical guarantees, would deliver a solution that does not suffer from the shortcomings stemming from the ANNS bounds: large approximation ratio and super-exponential dependence of the running time on the approximation ratio. Evaluating the practical applicability of such a method would be an interesting topic for further research.

---

> > ### Author Rebuttal · Reviewer_f7MX · 2026-04-03
> >
> > Thank you for the rebuttal. My main concerns have been addressed. Accordingly, I am increasing my overall score.

---

### Decision · Program_Chairs · 2026-04-30

**Decision:**

Accept (regular)

**Comment:**

This paper presents the first efficient dynamic algorithm for non-uniform facility location in high-dimensional Euclidean spaces, achieving a γ-approximation, O(log m) amortized recourse, and poly(d)·(m+n)^{O(1/γ)} amortized update time for sufficiently large constant γ.

The scores are: 4 (Weak accept), 4 (Weak accept), 5 (Accept), 5 (Accept) — overall positive.

In the rebuttal, all four reviewers acknowledged the authors' responses; three had their concerns fully resolved and maintained or increased their scores, while the fourth confirmed that all issues were adequately addressed and kept their positive evaluation.

The discussion was constructive and unanimous in supporting acceptance. Therefore, the recommendation is accept.